# Thalamus and claustrum control parallel layer 1 circuits in retrosplenial cortex

**Ellen KW Brennan[1,2†], Izabela Jedrasiak-Cape[1†], Sameer Kailasa[3†], Sharena P Rice[1,2], Shyam Kumar Sudhakar[1], Omar J Ahmed[1,2,4,5,6]\***

[1]Department of Psychology, University of Michigan, Ann Arbor, United States; [2]Neuroscience Graduate Program, University of Michigan, Ann Arbor, United States; [3]Department of Mathematics, University of Michigan, Ann Arbor, United States; [4]Michigan Center for Integrative Research in Critical Care, University of Michigan, Ann Arbor, United States; [5]Kresge Hearing Research Institute, University of Michigan, Ann Arbor, United States; [6]Department of Biomedical Engineering, University of Michigan, Ann Arbor, United States

**Abstract** The granular retrosplenial cortex (RSG) is critical for both spatial and non-spatial behaviors, but the underlying neural codes remain poorly understood. Here, we use optogenetic circuit mapping in mice to reveal a double dissociation that allows parallel circuits in superficial RSG to process disparate inputs. The anterior thalamus and dorsal subiculum, sources of spatial information, strongly and selectively recruit small low-rheobase (LR) pyramidal cells in RSG. In contrast, neighboring regular-spiking (RS) cells are preferentially controlled by claustral and anterior cingulate inputs, sources of mostly non-spatial information. Precise sublaminar axonal and dendritic arborization within RSG layer 1, in particular, permits this parallel processing. Observed thalamocortical synaptic dynamics enable computational models of LR neurons to compute the speed of head rotation, despite receiving head direction inputs that do not explicitly encode speed. Thus, parallel input streams identify a distinct principal neuronal subtype ideally positioned to support spatial orientation computations in the RSG.

**\*For correspondence:**
ojahmed@umich.edu

[†]These authors contributed equally to this work

**Competing interests:** The authors declare that no competing interests exist.

## Introduction

Activity in the granular retrosplenial cortex (RSG) is correlated with a wide variety of behaviors, including spatial navigation, learning, memory, fear conditioning, imagination, and planning for the future (*Alexander et al., 2020*; *Alexander and Nitz, 2017*; *Chang et al., 2020*; *Chrastil, 2018*; *Hinman et al., 2018*; *Mao et al., 2018*; *Miller et al., 2021*; *Miller et al., 2019*). The RSG is also among the most densely connected regions of the brain, integrating inputs from a bevy of cortical and subcortical sources and serving as part of the default mode network (*Greicius et al., 2009*; *Kaboodvand et al., 2018*; *Liu et al., 2019*; *Whitesell et al., 2021*). While these anatomical connections are well-documented (*Van Groen and Wyss, 2003*; *van Groen and Wyss, 1990*; *Whitesell et al., 2021*; *Wyss and Van Groen, 1992*), few studies have examined their functional nature (*Nitzan et al., 2020*; *Yamawaki et al., 2019b*; *Yamawaki et al., 2019a*; *Yamawaki et al., 2016a*), and cell-type specificity has yet to be sufficiently explored. Such knowledge is critical to develop a mechanistic understanding of how the RSG integrates information from multiple sources to carry out its spatial and non-spatial functions.

Cortical regions, including the RSG, receive inputs from one or more thalamic nuclei (*Herkenham, 1986*; *Herkenham, 1980*; *Jones, 2001*; *Jones, 1998*; *Peters, 1979*; *Van Groen and Wyss, 1995*). Thalamic relay cells (*Clascá et al., 2012*) are grouped into three distinct classes: core, intralaminar, and paralaminar/ventral midline (matrix) nuclei (*Hanbery and Jasper, 1953*; *Herkenham, 1986*; *Jones, 2001*; *Jones, 1998*; *Morison and Dempsey, 1941*; *Rubio-Garrido et al., 2009*).

**eLife digest** Sitting in your car, about to drive home after a long day at work, you realize you have no idea which way to go: you recognize where you are right now, and you remember the name of the street your house is on, but you cannot figure out how to get there. This spatial disorientation happens to people with damage to a brain region called the retrosplenial cortex, whose role and inner workings remain poorly understood. Recent evidence has shown that this area contains 'low-rheobase' neurons which are not seen anywhere else in the brain, but what do these neurons do?

Brennan, Jedrasiak-Cape, Kailasa et al. decided to explore the role of these neurons, focusing on the brain regions they are connected to. Experiments were conducted in mice using optogenetics, a technique that activates neurons using pulses of light. This revealed that brain areas involved in processing information about direction and position preferentially communicate with low-rheobase neurons rather than with nearby, more standard neurons in the retrosplenial cortex. The way these spatial signals are sent to the low-rheobase neurons allows these cells to 'calculate' how fast a mouse is turning its head using only information about which direction the mouse is facing. Essentially, this neuron can turn directional compass-like signals into a gyroscope signal that can track both direction and speed of head movement. These unique neurons may therefore be ideally suited to combine information about direction and space, suggesting that they may have evolved specifically to support spatial navigation.

Individuals with Alzheimer's disease show exactly the same type of spatial disorientation as individuals with direct damage to the retrosplenial cortex. This region is also one of the first to show altered activity in Alzheimer's disease. Exploring whether these unique retrosplenial neurons and their communication patterns are altered in Alzheimer's disease models could help to understand and potentially treat this debilitating condition.

Thalamic matrix nuclei project predominantly to layer 1 of many cortical regions (*Herkenham, 1986*; *Herkenham, 1980*), resulting in subtype-specific activation of cortical inhibitory (*Anastasiades et al., 2021*; *Cruikshank et al., 2012*; *Delevich et al., 2015*) and excitatory neurons (*Collins et al., 2018*; *Cruikshank et al., 2012*; *Guo et al., 2018*; *Rodriguez-Moreno et al., 2020*; *Rubio-Garrido et al., 2009*; *Van der Werf et al., 2002*; *Yamawaki et al., 2019b*). Matrix thalamo-cortical (TC) inputs from the anterior thalamus, which contains the highest proportion of head direction cells in the brain (*Taube and Bassett, 2003*), influence layer 5 pyramidal cells in RSG (*Yamawaki et al., 2019b*). However, potential subtype-specific responses and their functional impact on the encoding of directional information in RSG have yet to be examined.

The cortex, including the RSG, also receives widespread inputs from the densely connected claustrum (*Brown et al., 2017*; *Crick and Koch, 2005*; *Goll et al., 2015*; *Jackson et al., 2018*; *Kim et al., 2016*; *Narikiyo et al., 2020*; *Wang et al., 2017*; *White and Mathur, 2018*). As with TC signaling, the claustrum innervates both excitatory (*da Costa et al., 2010*; *Narikiyo et al., 2020*) and inhibitory (*Jackson et al., 2018*; *Narikiyo et al., 2020*; *Salerno et al., 1984*) neurons via precise laminar targeting (*Wang et al., 2017*) and projects to many of the same cortical regions as the thalamus (*Burman et al., 2011*). Claustrocortical (ClaC) projections are thought to regulate complex functions (*Crick and Koch, 2005*; *Goll et al., 2015*; *Jackson et al., 2018*; *Kitanishi and Matsuo, 2017*; *Narikiyo et al., 2020*; *Renouard et al., 2015*; *Smith et al., 2012*; *White and Mathur, 2018*), including the coordination and amplification of correlated signals across cortical regions (*Kim et al., 2016*; *Smith et al., 2012*; *Smythies et al., 2012*). Surprisingly, the RSG is one of few regions that does not return reciprocal connections to the claustrum (*Zingg et al., 2018*). Despite the unique unidirectional nature of this connection, no studies have investigated the functional nature or subtype-specificity of ClaC inputs to RSG neurons.

Each cortical region also receives inputs from select other cortical areas. The RSG receives direct innervation from the secondary motor cortex (*Yamawaki et al., 2016a*), contralateral RSG (*Sempere-Ferràndez et al., 2018*; *Wyss et al., 1990*), and anterior cingulate cortex (ACC; *Van Groen and Wyss, 2003*; *van Groen and Wyss, 1990*), among others. The RSG and ACC, in particular, have been implicated in several shared limbic and cognitive functions, including fear conditioning

(*Frankland et al., 2004*; *Han et al., 2003*; *Yamawaki et al., 2019a*; *Yamawaki et al., 2019b*), but the cellular targets of ACC inputs to RSG have not been identified.

Thalamic, claustral, and cortical inputs all converge in the RSG, but whether they do so via parallel or integrative control of target excitatory neurons remains unknown. Within the RSG, the superficial layers contain two subtypes of principal excitatory pyramidal neurons that have strikingly distinct physiology, morphology, and computational capabilities: the low-rheobase (LR) neuron and the regular-spiking (RS) neuron (*Brennan et al., 2020*; *Kurotani et al., 2013*; *Yousuf et al., 2020*). Here, we use channelrhodopsin (ChR2)-assisted circuit mapping (CRACM) to study the subtype-specific circuits formed by anterior thalamic, claustral, anterior cingulate, and dorsal subicular inputs to the RSG. Our results reveal a double dissociation in the precise organization of afferent axons and principal cell dendrites that can support parallel processing in superficial RSG. Specifically, we find that anterior thalamus and dorsal subiculum preferentially control LR neurons, whereas claustrum and anterior cingulate control RS cells. We show that the synaptic dynamics of thalamic inputs allow for the robust encoding of angular speed by LR cell models, even though these inputs only explicitly contain information about head direction, not speed. Together, our results suggest that LR neurons are ideally positioned to contribute to the initial spatial orientation computations performed in superficial RSG.

## Results

### Cell-type-specific thalamic control of layer three pyramidal cells in granular retrosplenial cortex

Layer 3 of the granular retrosplenial cortex (RSG) contains two types of principal pyramidal neurons: the low-rheobase (LR) neuron and the regular-spiking (RS) neuron (*Brennan et al., 2020*; *Kurotani et al., 2013*; *Yousuf et al., 2020*). Using whole-cell patch clamp recordings, we again distinguished LR from RS neurons by their unique intrinsic properties, predominantly the low rheobase, high input resistance, lack of adaptation, and narrow spike width (*Figure 1A–C*, *Figure 1—figure supplement 1*, *Supplementary file 1 - Table 1*, and *Supplementary file 1 - Table 2*; *Brennan et al., 2020*). We used channelrhodopsin-assisted circuit mapping (CRACM) to interrogate inputs to these two cell types. Consistent with previous studies (*Odagiri et al., 2011*; *Van Groen and Wyss, 2003*; *van Groen and Wyss, 1990*; *Yamawaki et al., 2019b*), virus injections into the anterodorsal and anteroventral thalamic nuclei (ADAV) resulted in ChR2-eYFP expression in thalamocortical (TC) axons and terminals predominantly in layer 1a (L1a; the most superficial part of layer 1) and in layer 3 (L3) of superficial RSG (*Figure 1D*). To test whether these thalamocortical inputs target the LR and RS neurons in layer 3, we stimulated the thalamic axons and terminals with 1 ms LED pulses at a 10 Hz frequency for 1 s at high LED power over the cell body of the patched neurons (*Figure 1E*; see Materials and methods). From resting membrane potentials of approximately −65 mV, LR neurons were strongly driven past spike threshold by these thalamic inputs while RS neurons exhibited far smaller EPSPs (*Figure 1F*). To quantify the impact of TC inputs on LR and RS spike rates, we next depolarized the cells until they were spiking at 10–25 Hz and delivered a single LED pulse (see Materials and methods). To rule out effects of feed-forward inhibition in these experiments, the GABA$_A$ antagonist, picrotoxin (50 µM), was added to the bath. Compared to their baseline activity, this stimulation evoked a significant increase in spiking in LR neurons (p=0.002, Wilcoxon rank sum test) but not RS neurons (p=0.19, Wilcoxon rank sum test; *Figure 1G*). The LR neurons thus had a significantly larger increase in spike rate evoked by the optical pulse than RS neurons (p=0.0012, Wilcoxon rank sum test; *Figure 1H*). This suggests that LR neurons receive stronger activation from TC inputs than their neighboring L3 RS neurons.

### Precise overlap of anterior thalamic afferents with LR, but not RS, apical dendrites

LR neurons are morphologically distinct from their neighboring RS neurons (*Brennan et al., 2020*; *Kurotani et al., 2013*; *Yousuf et al., 2020*). Given the precise laminar pattern of anterior thalamic inputs to RSG (*Figure 1D* and *Figure 2B*), we next investigated the overlap between anterior thalamic afferents and the dendritic morphologies of reconstructed LR (n = 10) and RS (n = 5) cells (see Materials and methods). Apical dendrites of LR neurons were most densely localized to L1a, with

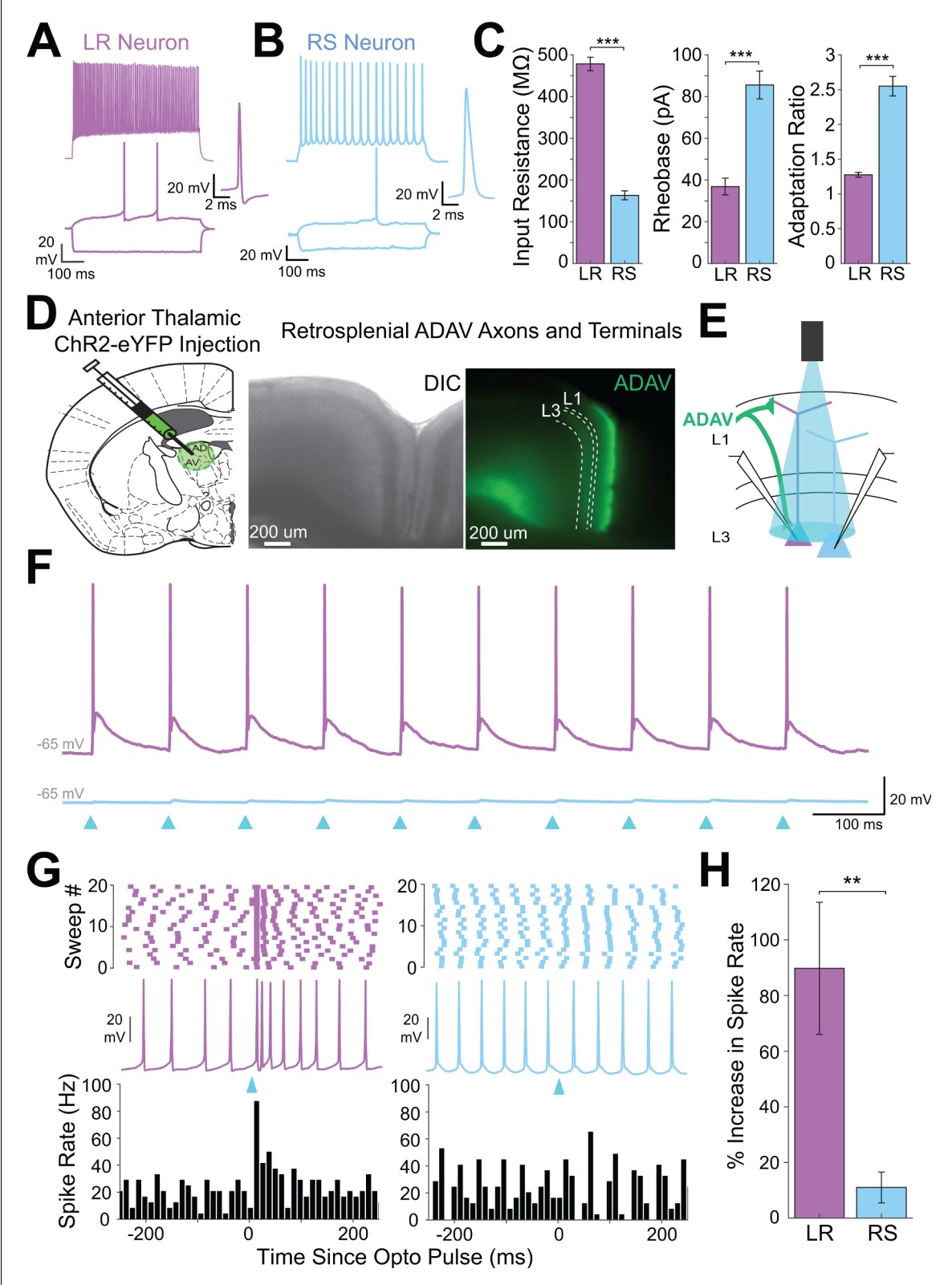

**Figure 1.** Anterior thalamic input controls LR, but not RS, neurons in superficial granular retrosplenial cortex. (**A**) Representative example of the firing properties of LR neurons. Top trace: sustained high-frequency firing of action potentials with little spike frequency adaptation at high current inputs in response to 200 pA current injection. Bottom trace: Delay to first spike at threshold current inputs (50 pA) and little to no sag when hyperpolarized (−100 pA). Right inset is a zoomed in view of the first spike in response to a near-threshold current input. Cell is held at −65 mV resting potential. (**B**)

*Figure 1 continued on next page*

*Figure 1 continued*

Same as A, but for a representative RS neuron. Note the spike frequency adaptation with high current inputs, delay to first spike at near-threshold current inputs, and sag potential seen with hyperpolarization. Cell is held at −65 mV resting potential. (**C**) Population data comparisons between LR and RS neurons of the following intrinsic properties: input resistance (p=1.31e-26; Wilcoxon rank sum), rheobase (p=1.24e-17; Wilcoxon rank sum), and adaptation ratio (p=2.94e-24; Wilcoxon rank sum). Error bars are standard error of the mean (SEM). (**D**) Left: Schematic showing the injection of optogenetic viral construct into the anterodorsal/anteroventral (ADAV) nuclei of the anterior thalamus. Middle: DIC image of RSG. Right: EYFP fluorescence image of RSG showing expressing anterior thalamic axons and terminal arbors. (**E**) Schematic of optogenetic stimulation of TC axons and terminal arbors over the patched cell body in RSG L3. (**F**) LR (purple) and RS (blue) cell responses to 10 Hz light pulses (see Materials and methods) at maximum LED power. LR cell shows spiking responses, while the RS cell has very small EPSPs. Both cells were held at −65 mV before stimulation. Blue triangles indicate light pulses. (**G**) Optogenetic activation of thalamic axons in RSG increases the firing rate of the LR (purple) but not RS (blue) neuron in a simultaneously patched pair. Neurons were held in ACSF +picrotoxin at a constant firing rate of 10–30 Hz via a 2 s current injection, and a 1 ms LED pulse was delivered 500 ms into the spike train (see Materials and methods). Top: Raster plots for all sweeps of a representative LR (purple) and RS (blue) example. Middle: Firing trace of one sweep for that same LR and RS pair. Bottom: PSTH plots evaluating all sweeps for the same LR and RS example cells. (**H**) Population analysis of the spike ratio for all tested LR (n = 7) and RS (n = 6) cells showing a significant increase in firing rate post-LED pulse in LR cells compared to RS cells (p=0.0012; Wilcoxon rank sum test). Error bars are SEM.     See *Figure 1—source code 1* and *Figure 1—source data 1* for MATLAB code and source data used to plot bar graphs in panel C and associated supplements.

The online version of this article includes the following source data, source code and figure supplement(s) for figure 1:

**Source data 1.** Cell-type specific intrinsic physiology.
**Source code 1.** Code to generate intrinsic physiology comparisons.
**Figure supplement 1.** Intrinsic properties of RSG layer 3 LR, layer 3 RS, and layer 5 RS neurons.

basal dendrites remaining confined to L3 (*Figure 2C*). In contrast, RS apical dendrites rarely entered L1a, instead existing in the deeper subdivisions of layer 1 (L1b/c), with basal dendrites primarily in upper layer 5 (L5; *Figure 2D*). Projection density analysis showed localization of ADAV axons and terminal arbors in L1a and weakly in L3 (*Figure 2C&D*). Thus, dendrites of LR, but not RS, cells selectively co-localize within layers 1a and 3 with ADAV TC afferents (*Figure 2C&D*). This anatomical colocalization of LR, but not RS, dendrites with TC axons coupled with the increase in LR, but not RS, spiking in response to TC optical stimulation indicates that the anterior thalamic-retrosplenial circuit at least partially obeys Peters' rule, which states that neuronal populations with anatomically overlapping axonal and dendritic arbors are more likely to show functional connectivity *Peters, 1979*; *Peters and Feldman, 1976*.

To further verify this, we examined the responses of L3 LR and RS neurons to ADAV input at L1a, where the highest projection density of ADAV inputs exist (see Methods). When delivering 1 ms high LED power light pulses, we indeed found that LR neurons had a significantly larger excitatory postsynaptic potential (EPSP) amplitude in response to the L1a LED stimulation compared to RS cell responses (p=0.0185, Wilcoxon rank sum test; *Figure 2E*). Notably, in response to layer one stimulation, LR cells also showed significantly larger EPSP amplitudes compared to L5 RS neurons (*Figure 2—figure supplement 1*), which have been previously shown to respond to TC input (*Yamawaki et al., 2019b*). This suggests that LR neurons, rather than layer 3 or 5 RS neurons, are the predominant RSG cell type receiving and processing head direction inputs from the anterior thalamus.

To further examine the effects of lamination of anterior thalamic inputs to LR and RS neurons on their resulting responses, we conducted a high-resolution protocol that used minimum thresholded LED power to optically stimulate every 20–30 μm along the neuron's longitudinal axis (*Figure 2F*; see Materials and methods). We found that LR neurons have the largest EPSP amplitude when the LED pulse is targeting layer 1a, corresponding with the peak projection density of ADAV axons and terminal arbors at this lamina. LR response amplitude then decreases as the LED moves away from the pia before reaching no response at the layer 1/2 border. As LED stimulation entered layer 3, the LR EPSP amplitude increased, consistent with the increase in projection density of the ADAV axons and terminal arbors, before again decreasing until their response amplitude reached 0 mV in layer 5 (*Figure 2F*). In contrast, the RS neuron showed much smaller EPSP amplitude with no significant variation in response across the layers (*Figure 2F&G*). To verify this functional sublaminar correlation between thalamocortical axons and pyramidal cell dendrites, we ran a population analysis comparing the EPSP amplitudes within LR and RS cell groups from all layers relative to L1/2, which represents our minimum projection density of ADAV inputs in the superficial layers. We found that within the LR

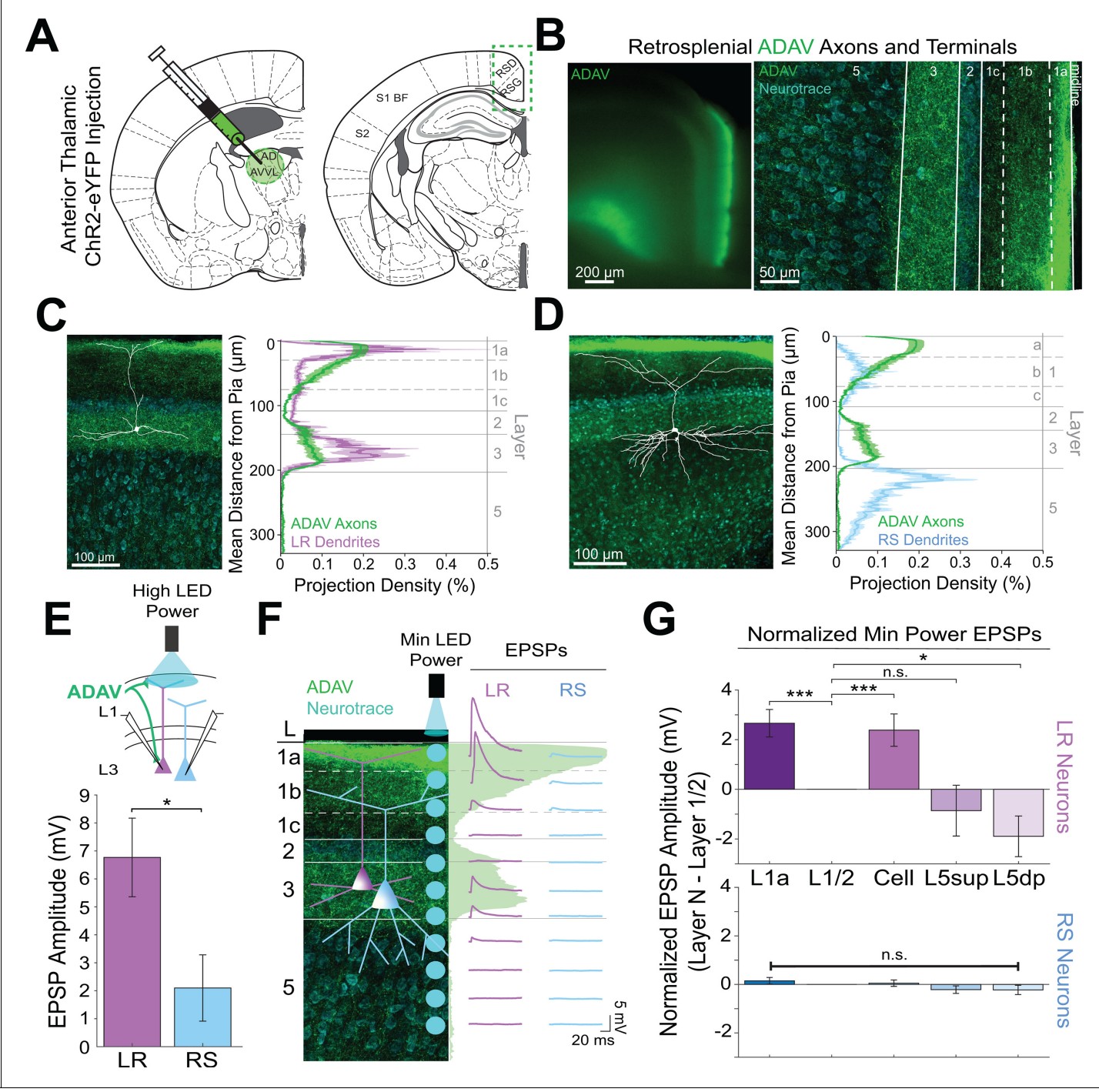

**Figure 2.** Thalamic control of LR neurons is explained by precise convergence of LR dendrites and anterior thalamic axons in layer 1a of granular retrosplenial cortex. (A) Schematic of channelrhodopsin injection into anterior thalamus (left) and RSG target recording region (right). (B) Left: epifluorescent image of the retrosplenial cortex showing expression of anterior thalamic axons and terminal arbors in green. Right: confocal image of layers 1–5 of RSG showing lamination of anterior thalamic axons and terminal arbors (green) and cell membrane marker, NeuroTrace (cyan). (C) Left: LR reconstruction superimposed on patch location in an RSG slice with thalamic projections (green) and NeuroTrace (cyan). Right: Projection density plot showing density of LR dendrites (n = 10; mean ± SEM shaded) in purple and density of ADAV axon expression (n = 5; mean ± SEM shaded) in green plotted as distance from the pia (μm). (D) Same as C for RS cells with dendrites (n = 5) plotted in blue. (E) Top: Schematic of L1a-targeted optogenetic stimulation. Bottom: Significantly larger EPSP amplitude at high LED power for LR (n = 15) cells compared to RS (n = 8) cells (p=0.0185; Wilcoxon rank sum test). Error bars are SEM. (F) Left: Confocal image of RSG slice with anterior thalamic axons and terminal arbors (green). Layers are demarcated, and schematic LR and RS neurons are placed in their representative locations within the superficial layers. Blue circles indicate targeting of minimum

*Figure 2 continued on next page*

*Figure 2 continued*

LED power stimulations, beginning at midline and extending into layer 5, stimulating every 20–30 µm (see Materials and methods). Right: Representative trace examples of an LR (purple) and RS (blue) neurons' responses to the LED stimulation at each location. Green shading is representative of the projection density of thalamic axons and terminal arbors across the shown layers. (**G**) Population analysis for LR and RS cells' normalized EPSP responses to minimum LED power optogenetic stimulations at L1a, L1/2 boundary, cell body in L3, superficial L5 (L5$_{sup}$), and deep L5 (L5$_{dp}$) locations. Top: LR cells have significantly larger responses at L1a (n = 58, p=0.00001, paired t-test) and cell body (n = 53, p=0.0006) compared to L1/2 (n = 59). Responses at L5$_{dp}$ (n = 10, p=0.0462) are significantly smaller than L1/2, and responses at L5$_{sup}$ (n = 12, p=0.420) do not significantly differ from L1/2. Bottom: EPSP amplitude in RS cells does not significantly differ at any stimulation location compared to L1/2 stimulation (L1a: n = 27, p=0.289; cell body: n = 25, p=0.702; L5$_{sup}$: n = 12, p=0.197; L5$_{dp}$: n = 11, p=0.261; L1/2: n = 27). Error bars are SEM. The paired t-test was used for all statistical comparisons. See *Figure 2—source data 1* for source data.

The online version of this article includes the following source data and figure supplement(s) for figure 2:

**Source data 1.** Laminar-specific responses to ADAV input.

**Figure supplement 1.** LR neuron responses to anterior thalamic and claustral inputs differ significantly from those of L5 RS cells.

cell population, LR cells have significantly larger responses at L1a (p=0.00001) and cell body (p=0.0006) compared to L1/2 (paired t-test; *Figure 2G*). This lamination of functional responses in the LR population precisely aligns with the lamination of ADAV axons and terminal arbors within these superficial layers. When stimulating in layer 5, there was no significant difference between EPSP amplitude at L1/2 and L5$_{sup}$ (p=0.420), while L5$_{dp}$ responses were significantly smaller than those at L1/2 (p=0.0462; *Figure 2G*), as would be expected given the complete lack of LR dendrites in L5. In contrast, RS EPSP amplitudes at layers 1a, cell body, L5$_{sup}$, and L5$_{dp}$ did not differ from responses at L1/2 (p>0.05 for all; paired t-test), indicating that there is no significant lamination of functional responses to thalamic inputs in the RS population, likely due to their relative lack of response to these inputs in general (*Figure 2G*). These relationships persisted when inhibition blocker picrotoxin (50 µM) or sodium channel blocker TTX (1 µM)+4 AP (100 µM) were added to the bath (data not shown; see Materials and methods). Thus, the connectivity between anterior thalamic TC arbors and L3 principal neurons in RSG highlights a circuit that depends on the precise sublaminar colocalization of both local principal cells' dendrites and incoming thalamocortical axons. This demonstrates that LR neurons are uniquely anatomically adapted to respond to incoming inputs from the anterior thalamus, where ~ 60% of neurons are head direction cells (*Taube, 1995*; *Taube and Bassett, 2003*), due to a precise overlap of their apical dendrites with the dense thalamic axons and terminals in the uppermost sublamina of L1 (L1a).

## Claustrum and anterior cingulate selectively control retrosplenial RS cells while avoiding LR cells

Two other main sources of input to RSG are the claustrum (CLA; *Van Groen and Wyss, 2003*; *van Groen and Wyss, 1990*) and anterior cingulate cortex (ACC; *Shibata and Naito, 2008*; *Van Groen and Wyss, 2003*; *van Groen and Wyss, 1990*). Both claustrocortical (ClaC) and ACC corticocortical (CC) projections have been anatomically shown to exhibit distinct laminar organization within its target cortical layers (*Van Groen and Wyss, 2003*; *Vogt and Miller, 1983*; *Wang et al., 2017*), but the precise functional targets of claustral and cingulate inputs to superficial RSG have, to our knowledge, never been examined. To address this, we next used CRACM, this time to examine CLA inputs to RSG LR and RS neurons and then compared these projections to RSG thalamic inputs. ChR2 injections into CLA (*Figure 3A*) resulted in expression of CLA axons and terminals in layers 1 c, 2, and 5 (*Figure 3B*). This expression pattern is distinct from the thalamic expression localized to layers 1a and 3 (*Figure 2B–D*). Dendritic lamination plots of LR and RS dendrites and projection density plots of CLA axons and terminal arbors show that RS, but not LR, apical dendrites anatomically overlap with CLA arbors in superficial layers 1 c and two and upper layer 5 (*Figure 3D*). Using the same targeted optogenetic stimulation approach as previously described (see Materials and methods), we found that RS neurons have significantly larger responses to CLA input compared to LR neurons when stimulated at layer 1/2 (p=0.0000035, Wilcoxon rank sum test; *Figure 3E*), the area of the strongest superficial projection density of CLA arbors. Similarly, layer 5 RS neurons exhibited significantly larger EPSP amplitudes in response to CLA inputs compared to LR cells (*Figure 2—figure supplement 1*). This again indicates that while LR neurons may be optimally

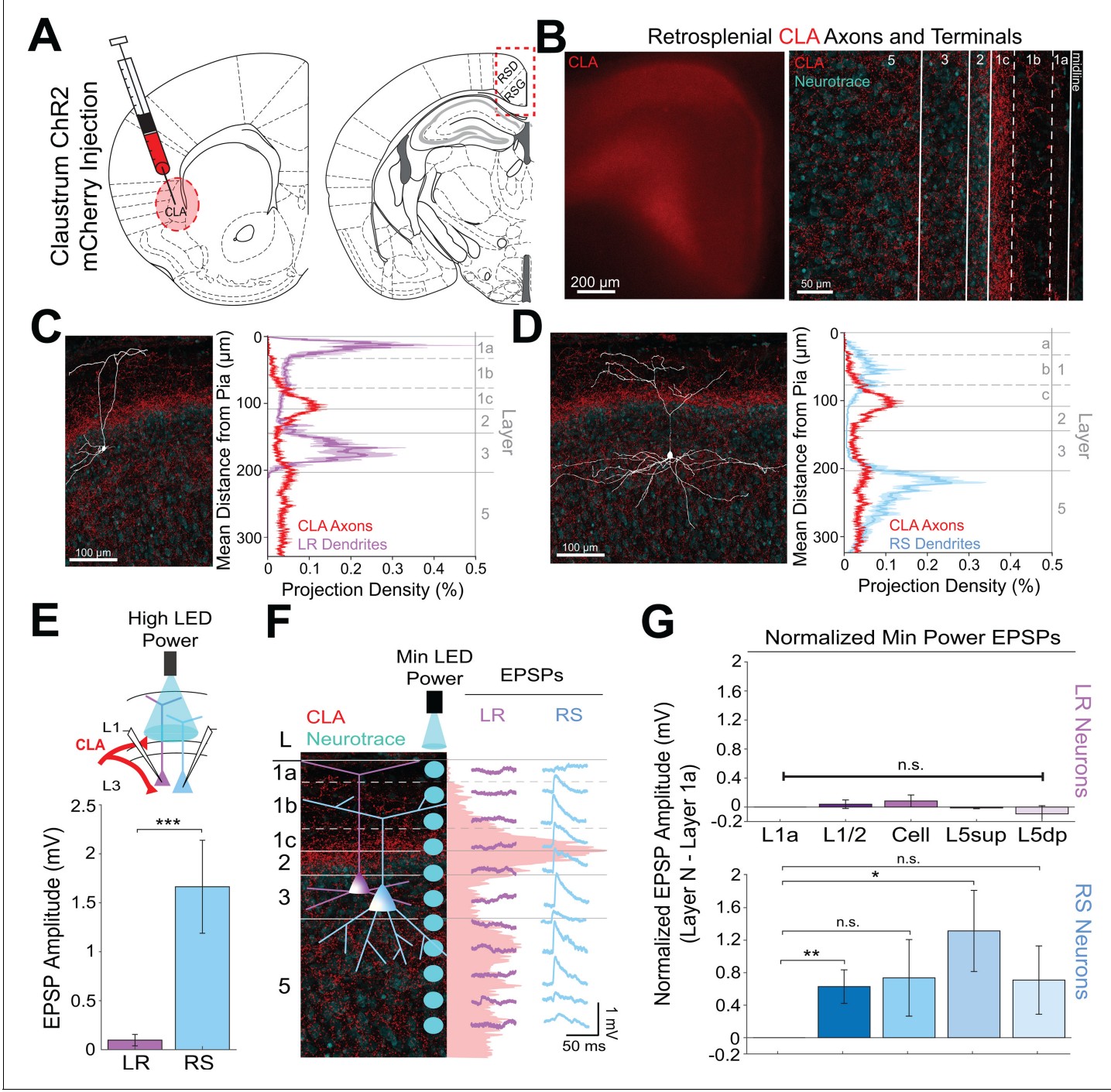

**Figure 3.** Claustral inputs to RSG selectively drive RS, not LR, neurons, consistent with stronger anatomical overlap of RS neuron dendrites with claustral axons. (A) Schematic of channelrhodopsin injection into claustrum (CLA; left) and target recording region, RSG (right; see Materials and methods). (B) Left: Epifluorescent image of the RSG brain slice showing expressing ClaC arbors in red. Right: confocal image of layers 1–5 of RSG showing lamination of claustral axons and terminal arbors (red) and NeuroTrace (cyan). (C) Left: LR reconstruction superimposed on patch location in an RSG slice with claustral projections (red) and NeuroTrace (cyan). Right: Projection density plot showing density of LR dendrites in purple (n = 10; mean ± SEM shaded) and density of CLA axon expression in red (n = 6; mean ± SEM shaded) plotted as distance from pia (μm). (D) Same as C for RS cells with dendrites (n = 5) plotted in blue. (E) Top: Schematic of L1/2-targeted optogenetic stimulation. Bottom: Significantly larger EPSP amplitude at high LED power for the RS (n = 16) cells compared to LR (n = 26) cells (p=0.0000035; Wilcoxon rank sum test). Error bars are SEM. (F) Left: Confocal image of RSG slice with CLA axons and terminal arbors (red) and NeuroTrace (cyan). Layers are demarcated, and schematic LR and RS neurons are placed in their representative locations within the superficial layers. Blue circles indicate targeting of the minimum LED power stimulations, beginning at midline and

*Figure 3 continued on next page*

Figure 3 continued

extending into layer 5, stimulating every 20–30 µm (see Materials and methods). Right: Representative trace examples of an LR (purple) and RS (blue) neurons' responses to the LED stimulation at each location. Red shading is representative of the projection density of CLA axons and terminal arbors across the layers. (G) Population analysis for LR and RS normalized EPSP responses to minimum LED power stimulations at L1a, L1/2, cell body, L5$_{sup}$, and L5$_{dp}$ locations. Top: EPSP amplitude in LR cells does not significantly differ at any stimulation location compared to L1a stimulation (L1/2: n = 24, p=0.5151; L3: n = 23, p=0.3276; L5$_{sup}$: n = 10, p=0.3434; L5$_{dp}$: n = 7, p=0.4348; L1a: n = 24). Bottom: RS cells have significantly larger responses at L1/2 (n = 18, p=0.00725) and L5$_{sup}$ (n = 9, p=0.0301) compared to L1a (n = 18), while normalized response amplitudes at cell body (n = 17, p=0.1377) and L5$_{dp}$ (n = 8, p=0.1355) did not differ from responses at L1a. The paired t-test was used for all statistical analyses. Error bars are SEM. See *Figure 3— source data 1* for source data.

The online version of this article includes the following source data for figure 3:

**Source data 1.** Laminar-specific responses to CLA input.

positioned to receive thalamic input, RS neurons in both RSG L3 and L5 are instead the primary recipients of CLA inputs.

To further examine the lamination of ClaC inputs to LR and RS neurons and their resulting responses, we again conducted the high-resolution CRACM protocol using minimum LED power (*Figure 3F*; see Materials and methods). We found that RS neurons have the largest EPSP amplitude when the LED pulse is stimulating both layer 1c and 5$_{sup}$ and weakest EPSP amplitude at layer 1a, corresponding with, respectively, the strongest and the weakest projection density of CLA afferents at these laminar locations. In contrast, the complementary LR cell had no response to LED stimulation at any of the laminar locations (*Figure 3F*). Population analysis showed that within the RS cell population, RS cells have significantly larger responses at L1/2 (p=0.00725) and L5$_{sup}$ (p=0.0301) compared to those at L1a (paired t-test; *Figure 3G*). This lamination of functional responses in the RS cell precisely aligns with the lamination of ClaC axons and terminal arbors within these layers. In contrast, LR EPSP amplitudes across all layers were much lower in magnitude and did not differ from responses at L1a (p>0.05 for all; paired t-test; *Figure 3G*), indicating no significant lamination of functional responses to ClaC inputs in the LR population. These relationships persisted when inhibition blocker picrotoxin (50 µM) or sodium channel blocker TTX (1 µM)+4 AP (100 µM) were added to the bath (data not shown; see Materials and methods).

Using the same CRACM approach to examine ACC inputs to RSG (*Figure 4A*), we found that corticocortical inputs from the anterior cingulate target L1b/c and, to a lesser extent, L5 (*Figure 4B*), partially resembling the laminar pattern seen with CLA arbors (*Figure 3B*) and overlapping precisely with RS, but not LR, dendrites (*Figure 3C&D*). Indeed, when stimulating ACC arbors at L1/2, RS cells had significantly larger EPSP amplitudes compared to LR cells (p=0.0019). When examining the lamination of these inputs with higher spatial resolution, we found that RS cells have the largest EPSP response to ACC inputs at L1/2, while the same minimum LED stimulation power elicits no response in layer 1a (*Figure 4F*). Population analysis of RS cells revealed significantly larger responses to optogenetic stimulation of ACC inputs at the L1/2 (p=0.000037) and cell body (p=0.000525) compared to L1a stimulation (*Figure 4G*), reflecting the precise overlap of ACC arbors and RS dendrites. In contrast, the much smaller LR EPSP amplitudes across all layers did not differ from responses at L1a (p>0.05 for all; paired t-test; *Figure 4G*). Again, the same results were seen when inhibition blocker picrotoxin (50 µM) or sodium channel blocker TTX (1 µM)+4 AP (100 µM) were added to the bath (data not shown; see Materials and methods). Thus, both claustrocortical and corticocortical inputs to RSG target RS apical dendrites at the lower divisions of L1 (L1c and L1b/c, respectively), whereas thalamocortical inputs preferentially target the apical dendrites of LR cells in upper L1 (L1a). This again indicates that parallel circuits in RSG process TC versus ClaC and CC information, and this parallel processing is enabled by precise sublaminar organization of afferent axons and layer three principal cell apical dendrites.

## Dorsal subiculum selectively controls LR, but not RS, cells

Previous work has shown that dorsal subiculum (DS), which serves to transmit allocentric spatial information such as axis cell and boundary vector signals (*Lever et al., 2009*; *Derdikman, 2009*; *Olson et al., 2017*; *Simonnet and Brecht, 2019*; *Bicanski and Burgess, 2020*), also precisely targets layer 2/3 principal neurons in RSG (*Nitzan et al., 2020*; *Yamawaki et al., 2019a*). These inputs overlap with LR cell bodies and basal dendrites and have been shown to evoke larger excitatory

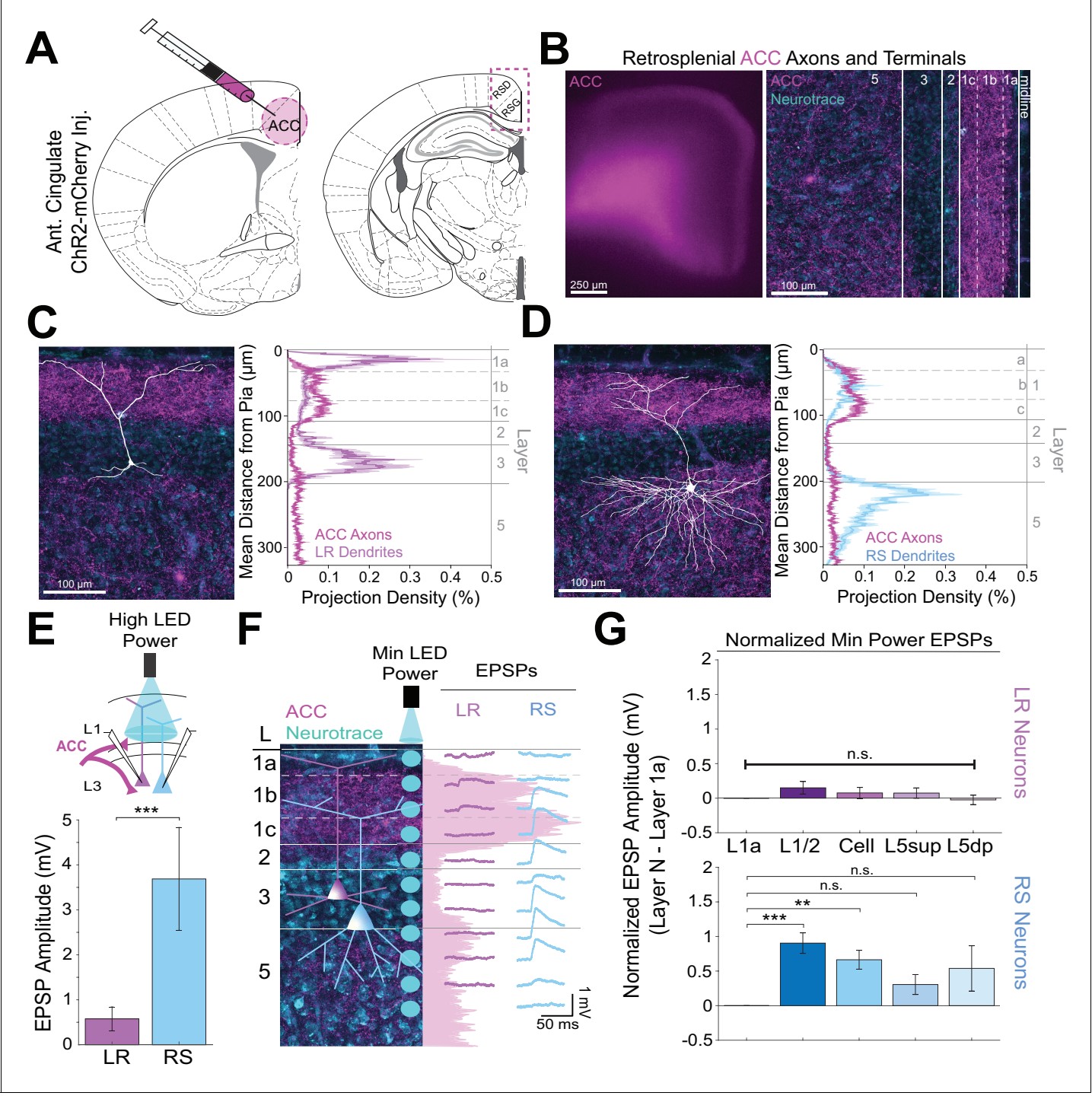

**Figure 4.** Anterior cingulate inputs to RSG selectively drive RS, not LR, neurons, consistent with anatomical overlap of RS neuron dendrites with anterior cingulate axons. (**A**) Schematic of channelrhodopsin injection into anterior cingulate (ACC; left) and target recording region, RSG (right; see Materials and methods). (**B**) Left: Epifluorescent image of the RSG brain slice showing expressing ACC arbors in magenta. Right: confocal image of layers 1–5 of RSG showing lamination of cingulate axons and terminal arbors (magenta) and NeuroTrace (cyan). (**C**) Left: LR reconstruction superimposed on patch location in an RSG slice with ACC projections (magenta) and NeuroTrace (cyan). Right: Projection density plot showing density of LR dendrites (n = 10; mean ± SEM shaded) in purple and density of ACC axon expression (n = 4; mean ± SEM shaded) in magenta plotted as distance from the pia (µm). (**D**) Same as C for RS cells with dendrites (n = 5) plotted in blue. (**E**) Top: Schematic of L1/2-targeted optogenetic stimulation. Bottom: Significantly larger EPSP amplitude at high LED power for the RS (n = 10) cells compared to LR (n = 16) cells (p=0.0019; Wilcoxon rank sum test). Error bars are SEM. (**F**) Left: Confocal image of RSG slice with ACC axons and terminal arbors (magenta) and NeuroTrace (cyan). Layers

*Figure 4 continued on next page*

Figure 4 continued

are demarcated, and schematic LR and RS neurons are placed in their representative locations within the superficial layers. Blue circles indicate targeting of the minimum LED power stimulations, beginning at midline and extending into layer 5, stimulating every 20–30 µm (see Materials and methods). Right: Representative trace examples of an LR (purple) and RS (blue) neurons' responses to the LED stimulation at each location. Magenta shading is representative of the projection density of ACC axons and terminal arbors across the layers. (G) Population analysis for LR and RS normalized EPSP responses to minimum LED power stimulations at L1a, L1/2, cell body, $L5_{sup}$, and $L5_{dp}$ locations. Top: EPSP amplitude in LR cells does not significantly differ at any stimulation location compared to L1a stimulation (L1/2: n = 19, p=0.1126; cell body: n = 19, p=0.35; $L5_{sup}$: n = 13, p=0.3423; $L5_{dp}$: n = 13, p=0.7166; L1a: n = 19). Bottom: RS cells have significantly larger responses at L1/2 (n = 14, p=0.000037) and cell body (n = 14, p=0.000525) compared to L1a (n = 14) and did not significantly differ at $L5_{sup}$ (n = 11, p=0.0607) or $L5_{dp}$ (n = 10, p=0.1340) compared to L1a. The paired t-test was used for all statistical analyses. Error bars are SEM. See *Figure 4—source data 1* for source data.

The online version of this article includes the following source data for figure 4:

**Source data 1.** Laminar-specific responses to ACC input.

---

postsynaptic currents in superficial compared to layer 5 pyramidal neurons (*Nitzan et al., 2020*; *Yamawaki et al., 2019a*). These results suggest that DS projections to RSC may be targeting LR cells, potentially resulting in converging spatial information from both ADAV and DS onto LR cells. However, it remains unknown whether LR or layer 3 RS cells are the predominant target of these DS projections. To investigate this, we repeated our CRACM protocols and examined the functional responses of LR and layer 3 RS cells to DS inputs (*Figure 5A&B*). We found that DS afferents to layer 3 of RSG precisely overlap with LR basal and, to a lesser degree, proximal apical dendrites (*Figure 5C*) but not RS basal dendrites (*Figure 5D*). Correspondingly, LR neurons responded with significantly larger EPSPs compared to RS neurons when DS inputs were stimulated at the cell body in layer 3 (p=0.004; *Figure 5E*). Our high-resolution protocol and population analysis also confirmed that LR cells had significantly larger responses at layers 1/2 and 3 compared to L1a (L1/2 p=0.0132, L3 p=0.0135; *Figure 5F&G*), suggesting the possibility of integration of synchronous anterior thalamic and dorsal subicular inputs by LR neurons. In contrast, RS cells showed no significant laminal differences in response to DS input (p>0.05 for all; *Figure 5G*). Thus, inputs from both the dorsal subiculum and anterior thalamus target LR cells, reflecting their precise overlap with LR dendrites, while the claustrum and anterior cingulate cortex instead target RS cells.

## Precise anatomical overlap of LR versus RS dendrites with distinct incoming axons facilitates parallel circuits in superficial RSG

We next extended our physiological examination of superficial RSG circuits to include correlational analyses and dual injection experiments (*Figure 6A*). We found that LR dendrites are strongly and significantly positively correlated with both ADAV (r = 0.88, p<0.001) and DS (r = 0.69, p=1.41e-254) axons. In contrast, RS dendrites are strongly correlated with CLA (r = 0.29, p=4.82e-36) and ACC (r = 0.43, p=5.73e-82) axons. Notably, LR dendrites are significantly negatively correlated with both CLA (r = −0.49, p=1.16e-110) and ACC (r = −0.15, p=1.00e-10) axons, and RS dendrites are negatively correlated with ADAV (r = −0.55, p=2.47e-145) and DS (r = −0.44, p=7.74e-85) axons (*Figure 6B&C*). These anatomical results mirror our functional findings that LR neurons are preferentially driven by both ADAV and DS inputs, while RS neurons are targeted by CLA and ACC inputs.

We also compared laminar expression of afferents from the four input regions and found that ADAV and DS inputs to RSG, which both at least in part target layer 3, were significantly positively correlated with one another (r = 0.55, p=2.93e-143). Similarly, CLA and ACC inputs, which both target the lower divisions of L1 as well as L5, were also significantly positively correlated (r = 0.23, p=2.3e-23). However, ADAV and DS inputs were significantly negatively correlated with both CLA and ACC inputs, confirming that these two streams target separate sublayers within RSG (*Figure 6B,C*). Dendrite-dendrite correlations of LR and RS cell populations were also strongly anti-correlated (r = −0.65, p=3.00e-215; *Figure 6C*), highlighting the distinct anatomical organization of these two neighboring principal cell types. Taken together, these axon-axon, dendrite-dendrite, and axon-dendrite correlations indicate that precise, fine-grained laminar organization of LR and RS dendrites and afferent axons creates a parallel circuit in which LR neurons are selectively optimized to integrate incoming spatial information (*Figure 6D*).

Supplemental examination of contralateral RSG (cRSG) inputs to RSG showed lamination and resulting cell-type-specific targeting resembling that of projections from CLA and ACC (*Figure 6—*

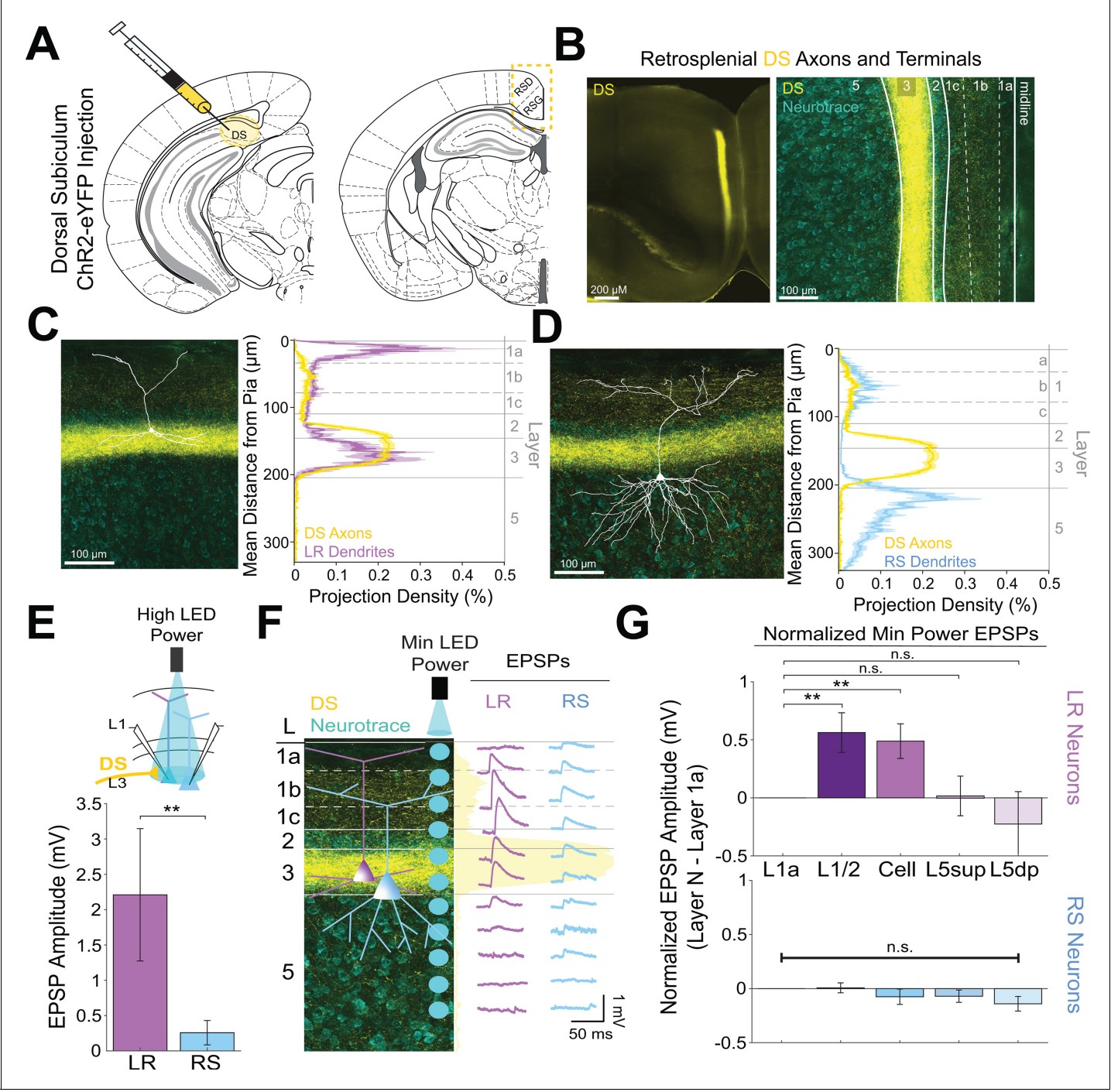

**Figure 5.** Dorsal subiculum inputs to RSG selectively drive LR, not RS, neurons, consistent with anatomical overlap of LR neuron dendrites with subicular axons. (**A**) Schematic of channelrhodopsin injection into dorsal subiculum (DS; left) and target recording region, RSG (right; see Materials and methods). (**B**) Left: Epifluorescent image of the RSG brain slice showing expressing DS arbors in yellow. Right: confocal image of layers 1–5 of RSG showing lamination of DS axons and terminal arbors (yellow) and NeuroTrace (cyan). (**C**) Left: LR reconstruction superimposed on patch location in an RSG slice with DS projections (yellow) and NeuroTrace (cyan). Right: Projection density plot showing density of LR dendrites (n = 10; mean ± SEM shaded) in purple and density of DS axon expression (n = 4; mean ± SEM shaded) in yellow plotted as distance from the pia (μm). (**D**) Same as C for RS cells with dendrites (n = 5) plotted in blue. (**E**) Top: Schematic of cell body-targeted optogenetic stimulation at high LED power (see Materials and methods) conducted in standard ACSF. Bottom: Bar graph showing significantly larger EPSP amplitude at high LED power for the LR (n = 9) cells compared to RS (n = 8) cells (p=0.004; Wilcoxon rank sum test). Error bars are SEM. (**F**) Left: Confocal image of RSG slice with DS axons and terminal arbors (yellow) and NeuroTrace (cyan). Layers are demarcated, and schematic LR and RS neurons are placed in their representative locations within the superficial layers. Blue circles indicate targeting of the minimum LED power stimulations, beginning at midline and extending into layer 5, stimulating every 20–30 μm

*Figure 5 continued on next page*

*Figure 5 continued*

(see Materials and methods). Right: Representative trace examples of an LR (purple) and RS (blue) neurons' responses to the LED stimulation at each location. Yellow shading is representative of the projection density of DS axons and terminal arbors across the layers. (G) Population analysis for LR and RS normalized EPSP responses to minimum LED power stimulations at L1a, L1/2, cell body, L5$_{sup}$, and L5$_{dp}$ locations. Top: LR cells have significantly larger responses at L1/2 (n = 8, p=0.0132) and cell body (n = 8, p=0.0135) compared to responses at L1a (n = 8), while responses at L5$_{sup}$ (n = 8, p=0.9146) and L5$_{dp}$ (n = 7, p=0.4902) do not differ from those at L1a. Bottom: EPSP amplitude in RS cells does not significantly differ at any stimulation location compared to L1a stimulation (L1/2: n = 7, p=0.8811; cell body: n = 7, p=0.3304; L5$_{sup}$: n = 7, p=0.2260; L5$_{dp}$: n = 6, p=0.0816; L1a: n = 7). The paired t-test was used for all statistical analyses. Error bars are SEM. See *Figure 5—source data 1* for source data.

The online version of this article includes the following source data for figure 5:

**Source data 1.** Laminar-specific responses to DS input.

*figure supplement 1*). Specifically, cRSG inputs targeted L1c, L2, and L5, resulting in strong activation of RS cells, with almost no responses in LR cells (*Figure 6—figure supplement 1*). This again suggests that inputs from CLA, ACC and cRSG follow a structured pattern by which these inputs preferentially control RS cells, while ADAV and DS projections target LR cells. Importantly, laminar dichotomies similar to those seen here in RSG also exist between TC and ClaC/corticocortical projections to other cortical regions such as the medial prefrontal cortex (*Cruikshank et al., 2012*), as also revealed by our examination of anatomical datasets from the Allen Brain Institute (*Figure 6—figure supplement 2*). Thus, our findings may also highlight a more universal framework by which other cortical regions integrate and process thalamic versus claustral and cortical inputs.

## Anterior thalamic input to LR neurons is uniquely depressing

To investigate the functional implications of these parallel circuits, we next examined the short-term dynamics of each input onto the RSG principal cells. LR short-term dynamics were examined in response to ADAV and DS inputs, while RS short-term dynamics were examined in response to CLA and ACC inputs (see Materials and methods). In contrast to previously documented thalamocortical matrix inputs to both superficial pyramidal cells and interneurons (*Anastasiades et al., 2021*; *Cruikshank et al., 2012*), 10 Hz anterior thalamic inputs from ADAV to RSG LR neurons were uniquely and significantly depressing (p=1.02e-5, Wilcoxon rank sum test; *Figure 7A&B*). In contrast, 10 Hz DS inputs to LR neurons were not depressing and exhibited weak facilitation (*Figure 7A&B*). Both CLA and ACC inputs to RS cells were also weakly facilitating (*Figure 7C&D*). Thus, the synaptic depression of anterior thalamic inputs to LR neurons is unique among the inputs examined here and also distinct from anterior thalamic inputs to superficial principal neurons in PFC and ACC (*Cruikshank et al., 2012*). As expected, we found that ADAV inputs to LR cells do not show synaptic depression at the much slower stimulation frequency of 0.1 Hz (*Figure 7—figure supplement 1* panel C), but 40 Hz inputs resulted in stronger depression than 10 Hz (p=0.0022, Wilcoxon rank sum test; *Figure 7—figure supplement 1* panel B).

## Short-term depression of anterior thalamic inputs enables encoding of angular head speed by LR cells

We next used computational modeling to elucidate the functional role of synaptic depression in this circuit, especially with respect to the processing of head direction input (*Taube, 1995*; *Taube and Bassett, 2003*). Up to 60% of cells in the anterior thalamus are classical head direction (HD) cells (*Taube and Bassett, 2003*). Each HD cell has a unique preferred direction at which its firing rate is highest when the head is facing that direction. The preferred directions of all cells in the HD ensemble span the full range of compass directions (*Taube, 1998*). Our modeling setup consisted of a postsynaptic RSG LR cell receiving input from an ensemble of 7500 presynaptic HD cells via depressing synapses (for full details, see Materials and methods). In the initial simulations shown in *Figure 8*, the morphologically realistic model LR cell (*Brennan et al., 2020*) received uniform inputs (all HDs being equally likely) via depressing synapses with Tsodyks-Markram short-term dynamics (*Tsodyks et al., 1998*). The synaptic parameters were fit to match the experimentally observed response of LR cells to optogenetic stimulation of thalamic afferents reported above (*Figure 7*).

We found that the firing rate of the postsynaptic LR cell receiving HD input via depressing synapses was strongly correlated with angular head speed, giving rise to symmetrical angular head velocity tuning (*Figure 8B&C*). Identical HD inputs transmitted via non-depressing synapses resulted in LR

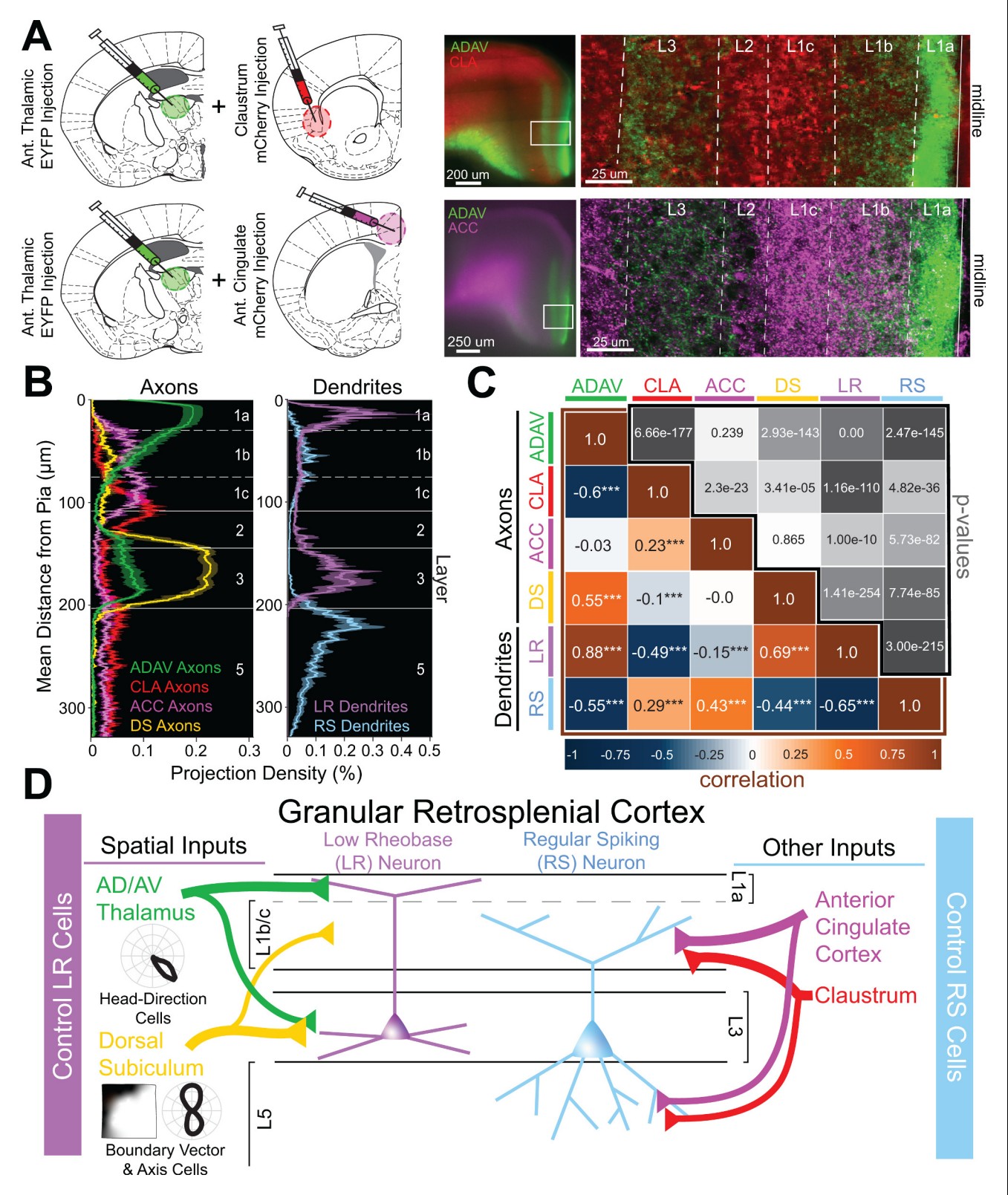

**Figure 6.** Two parallel circuits in superficial granular retrosplenial cortex. (**A**) Left: Schematics of dual injections into anterior thalamus (ADAV) and claustrum (CLA; top) or anterior cingulate (ACC; bottom). Right: Resulting dual expression of ADAV (green) and CLA (red; top) or ACC (magenta; bottom) axons with zoomed-in confocal view of layers, demarcated by white lines. (**B**) Left: Projection density of anterior thalamus (ADAV; green; n = 5), claustrum (CLA; red; n = 6), anterior cingulate (ACC: magenta; n = 4), and dorsal subiculum (DS; yellow; n = 4; mean ± SEM shaded for all) axons and

*Figure 6 continued on next page*

*Figure 6 continued*

terminal arbors in RSG. Note distinct sublaminar distribution of axons from ADAV, ACC, and CLA in layer 1. Right: Projection density of LR (purple; n = 10) and RS (blue; n = 5; mean ± SEM shaded for all) neurons. Note the distinct difference between LR and RS dendrite lamination across the layers. (C) Correlation matrix of means for all axon-axon, axon-dendrite, and dendrite-dendrite comparisons. Note that LR dendrites are significantly positively correlated with ADAV and DS axons but negatively correlated with CLA and ACC axons, while RS dendrites are significantly positively correlated with CLA and ACC axons but negatively correlated with ADAV and DS axons. (D) Summary schematic showing selective control of LR neurons by inputs from ADAV and DS, including head direction signals. In contrast, RS neurons are preferentially controlled by inputs from the CLA and ACC. This precise organization of principal cell dendrites and afferent axons forms two parallel circuits in superficial RSG.

The online version of this article includes the following figure supplement(s) for figure 6:

**Figure supplement 1.** Contralateral RSG projections drive RS, but not LR, neurons.
**Figure supplement 2.** Sublaminar differences in thalamic vs claustral projections to medial prefrontal cortex.

firing rates that were uncorrelated with angular head speed (*Figure 8B,E–F*). It is important to note that in our model, firing rates of input HD cells were not explicitly modulated by angular head velocity; therefore, the observed speed tuning resulted strictly from the depressing synaptic dynamics. For the parameter set whose results are depicted, LR cell firing rate was optimally correlated with head speed 24 ms in the past, confirmed by both cross-correlation and mutual information analyses (*Figure 8E&F*). We obtained the same results when modeling an ensemble of only 2500 presynaptic HD cells, indicating that our results are robust to the size of the presynaptic population (*Figure 8— figure supplement 1*). Taken together, these results suggest that synaptic depression of HD ensemble inputs introduces an angular head speed signal into the LR population, producing neurons that are more responsive during faster head turns, and potentially supporting the spatial orientation encoding functions attributed to the RSG (*Epstein, 2008*; *Ino et al., 2007*; *Milczarek et al., 2018*; *Miller et al., 2019*).

## Anticipatory firing of thalamic head direction cells improves postsynaptic speed encoding by retrosplenial LR neurons

Anterior thalamic HD cells display anticipatory firing (*Blair et al., 1997*), a phenomenon where an HD cell becomes most active at a fixed time interval before the animal is facing that cell's preferred direction (*Figure 9A*). This temporal offset associated with each cell is called its anticipatory time interval (ATI). In anterior thalamus, the mean ATI has been reported as 25–50 ms (*Taube, 2010*). In our initial simulations, shown in *Figure 8*, we drew the ATI of each HD cell randomly from distributions matching these known in vivo ranges. To more systematically understand if and how the ATI of HD cells influences angular head speed coding by LR neurons, we next performed a series of simulations using various fixed ATI values for the entire HD population (*Figure 9A*) and analyzed the resulting postsynaptic LR response.

Increasing ATI from 0 ms (no anticipation by the HD cell) to higher values (HD cell firing prior to facing the preferred HD) improved the lag between LR firing rate and angular head speed, with larger ATIs resulting in shorter latency between the head movement and LR coding of that angular head speed (*Figure 9C*). Remarkably, larger ATIs also improved the angular head speed tuning of LR neurons independent of lag, as quantified by the maximum value (over the full range of lags) of cross-correlation or cross-mutual information (*Figure 9B*). Thus, anticipatory firing of HD cells may constitute a powerful coding principle in the thalamo-retrosplenial circuit, helping LR cells to not only encode the current head speed with minimal lag, but also better encode the head speed independent of the lag.

In order to better understand this surprising effect, as well as how speed tuning generally arises, we analytically studied a simplified mean-field model of this thalamo-retrosplenial circuit (*Figure 9— figure supplement 1*). Our analysis allowed us to mathematically derive that LR activity should encode head speed with depressing synapses from simplified HD cells (see *Appendix*). Moreover, the analysis showed that anticipatory firing compensates for the lag introduced by integration time, leading to a theoretical parameter regime for which postsynaptic speed coding can be essentially perfect (where LR firing rate reflects head speed exactly with minimal latency). The observed improvement in quality of speed coding with increases in ATI corresponds to moving closer to this regime. Finally, our analysis showed that postsynaptic activity, at least at lower rotational speeds, is proportional to the square of head speed, thereby explaining the concave-up, parabolic shape of

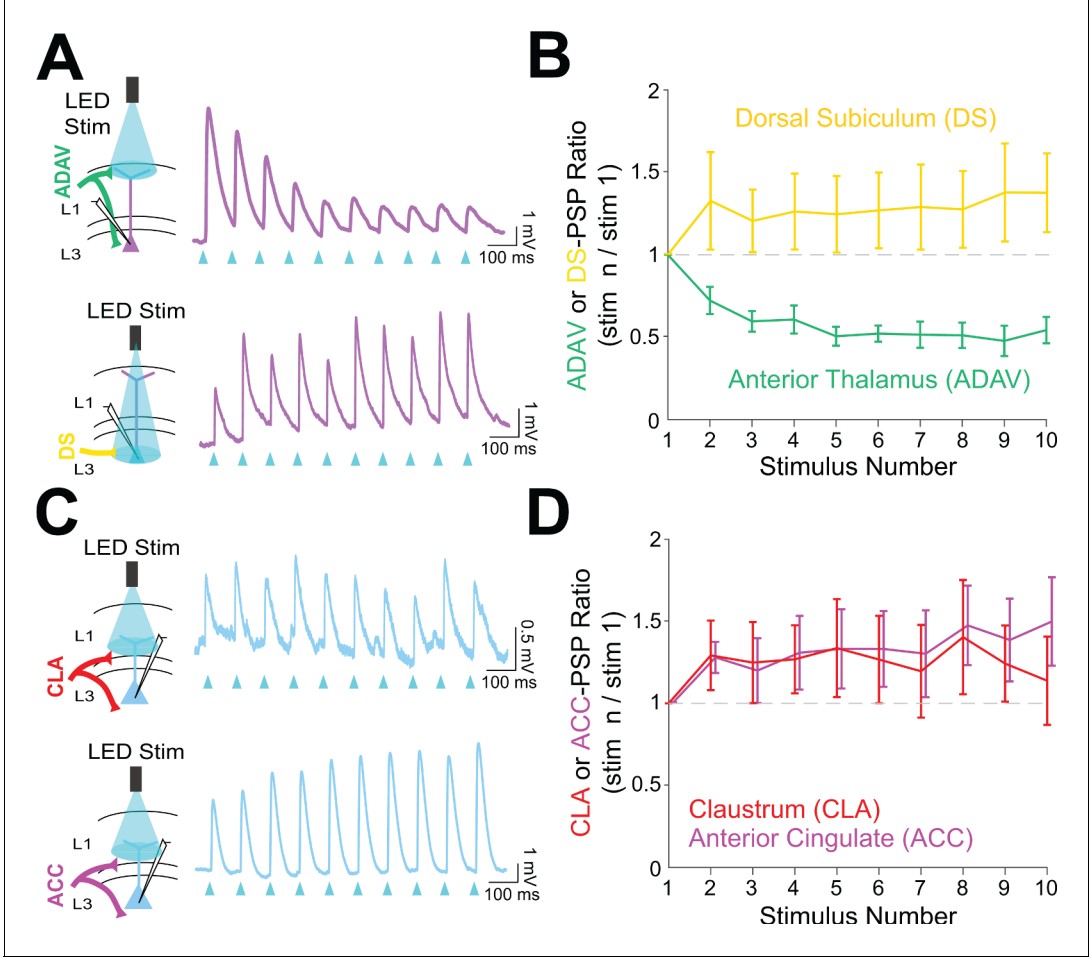

**Figure 7.** Anterior thalamic inputs to LR neurons evoke robust synaptic depression. (**A**) Top: Schematic showing anterior thalamic (ADAV) inputs to an LR neuron being optically stimulated at L1a (left). Example trace from a representative LR cell in response to 10 Hz ADAV stimulation shows clear synaptic depression (purple). Blue triangles represent light pulses (right). Bottom: As above, but now for dorsal subicular (DS) input to an LR cell showing that LR responses to DS synaptic inputs are weakly facilitating. (**B**) Group synaptic dynamics for LR neurons in response to ADAV (green; n = 9) and DS (yellow; n = 9) inputs. ADAV synapses are strongly depressing, while DS synapses are weakly facilitating. (**C**) Top: Schematic showing claustral (CLA) inputs to an RS neuron being optically stimulated at L1/2 (left). Example trace from a representative RS cell in response to 10 Hz CLA stimulation shows weak synaptic facilitation (blue). Blue triangles represent light pulses (right). Bottom: As above, but now for anterior cingulate (ACC) input to an RS cell showing that RS responses to ACC synaptic inputs are weakly facilitating. (**D**) Group synaptic dynamics for RS neurons in response to CLA (red; n = 8) and ACC (magenta; n = 7) inputs. Both CLA and ACC synapses are weakly facilitating. See *Figure 7—source data 1* for source data.

The online version of this article includes the following source data and figure supplement(s) for figure 7:

**Source data 1.** Short-term dynamics of postsynaptic responses.

**Figure supplement 1.** Anterior thalamic inputs to LR neurons are strongly depressing at higher frequencies.

the speed-firing rate curve (*Figure 8C* and *Figure 8—figure supplement 2*). Thus, both the simulations and the analytical theory show that synaptic depression of HD cell input onto LR cells should result in angular speed coding in the LR cell firing rate, and that anticipatory firing improves both the precision and quality of this coding.

## Non-uniform HD inputs can allow LR cells to encode both head direction and speed, with a tradeoff

In the simulations above, we utilized a uniform distribution of HD inputs to each LR cell, such that each preferred direction was equally represented. However, in practice, it is likely that there will be heterogeneity in the preferred HDs of the cells providing inputs to any given retrosplenial LR cell. To study the extent to which speed coding persists under these conditions, we simulated a population

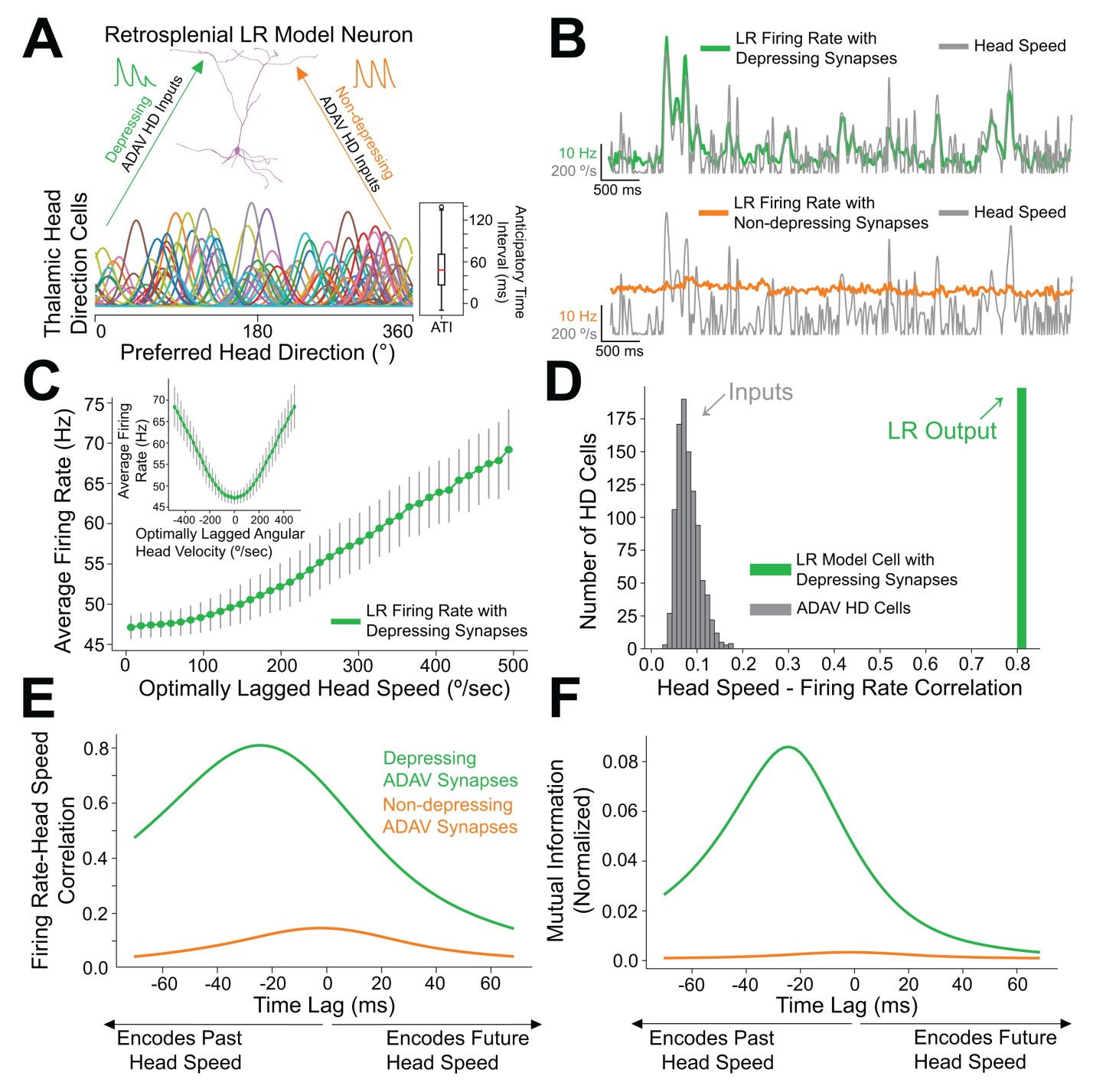

**Figure 8.** Depressing thalamocortical synapses allow LR cells to compute head speed from directional inputs. (**A**) We modeled a heterogeneous population of HD cells providing input to a single LR neuron via either depressing (green) or non-depressing (orange) synapses. Presynaptic HD cells varied in their tuning width, maximum firing rate, background firing rate, and anticipatory time interval (ATI). Tuning curves of a randomly selected subset of 100 (out of 7500) HD cells in the simulated ensemble are depicted here. Boxplot on the right depicts the empirical distribution of ATIs of all HD cells in the presynaptic population, with the mean ATI chosen as 50 ms in accordance with **Taube, 2010**. (**B**) Sample traces spanning ten seconds of simulation time. Green: firing rate of postsynaptic LR cell receiving HD input via depressing synapses; Orange: firing rate of postsynaptic LR cell receiving HD input via non-depressing synapses; Gray: head turning speed. Note that for the non-depressing inputs, the firing rate of the LR cell remains approximately constant throughout. In contrast, depressing synapses produce a firing rate whose fluctuations visibly reflect fluctuations in the angular head speed. (**C**) Firing rate of postsynaptic LR cell plotted against head speed 24 ms in the past shows a clear monotonic relationship. A similar relationship exists between firing rate and current angular head speed (see panels **E**, **F**). The inset shows firing rate as a function of angular head

*Figure 8 continued on next page*

*Figure 8 continued*

velocity. The error bars indicate the standard deviation of all firing rates in each AHV bin; note that these variabilities are low, emphasizing the consistency and reliability of this code. (D) The gray histogram depicts the correlations between input HD cell firing rate and angular head speed during active (>10 Hz firing) windows. Since presynaptic firing rate was only explicitly modulated by head direction and not head speed, these values are low, with a mean value of 0.08. The green bar depicts correlation between postsynaptic firing rate and head speed at the optimal time lag (24 ms, see panels E, F). Hence, LR cells can utilize depressing inputs from HD cells to compute de novo head speed. (E) Cross-correlation between LR cell firing rate and head speed. We computed the Pearson correlation between the firing rate of the postsynaptic LR cell and the head speed L ms in the future for varying values of L, ranging from −70 to 70 ms. Postsynaptic firing rate was maximally correlated with head speed 24 ms in the past. (F) Same as E but for cross-mutual information. Cross-mutual information identifies the same optimal lag as cross-correlation.

The online version of this article includes the following figure supplement(s) for figure 8:

**Figure supplement 1.** Angular head speed coding by LR cells is robust regardless of presynaptic thalamic head direction cell count.

**Figure supplement 2.** LR firing rate exhibits approximately quadratic scaling for low speeds.

of 200 LR cells, each of which received inputs from a randomly selected distribution of preferred directions for the presynaptic HD cells (*Figure 10Ai,,Bi,Ci*). We then analyzed the information content of each LR cell's firing rate, assessing the correlation and mutual information with both head direction and speed. We found that single cell activity in the LR population could encode both head direction and head speed (*Figure 10Aii,Bii,Cii*), but there was a clear tradeoff between the two types of information.

This tradeoff between directional versus speed encoding was explained by the level of heterogeneity in the presynaptic preferred direction distribution (*Figure 10D–F*), making intuitive sense. If an LR cell receives inputs that represent each HD equally, then it is not biased towards any one HD and encodes direction poorly. However, it encodes head speed very strongly in this case because it is able to sample the speed across all directions—there are no directional blind spots where the LR cell

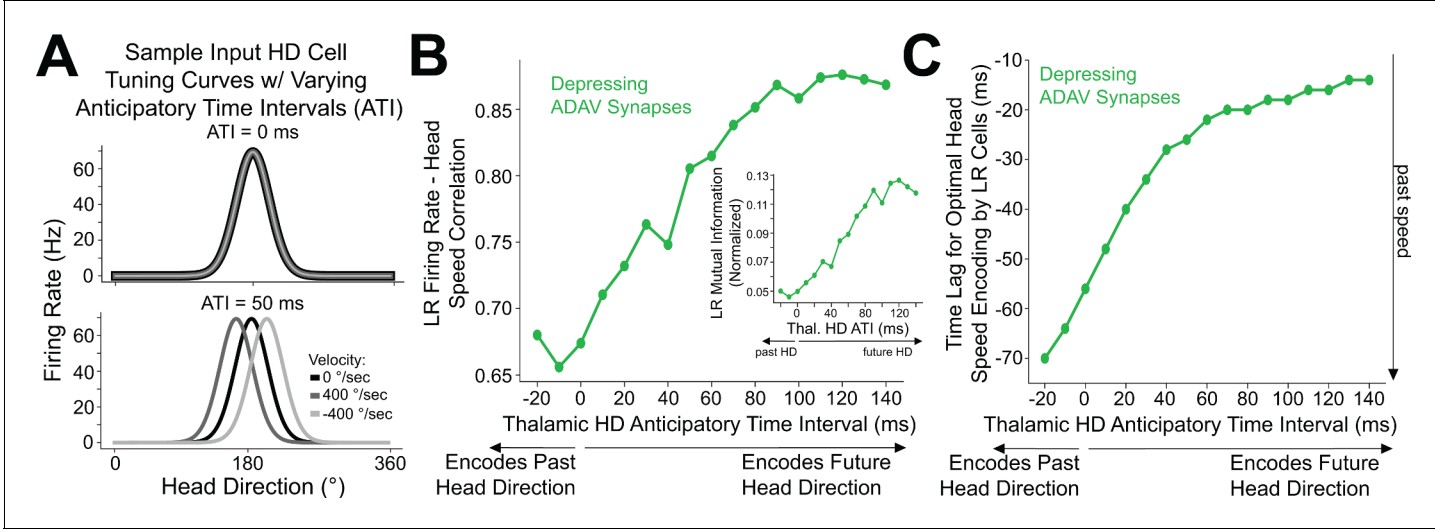

**Figure 9.** Encoding of future head direction in thalamus helps to better encode present head speed in retrosplenial LR cells. (A) The schematic depicts tuning curves for an HD cell with a preferred direction of 180 degrees. Top: the tuning curves of this cell if it displayed no anticipatory firing (ATI = 0 ms). Note that clockwise and counterclockwise turns produce identical tuning curves in this case. Bottom: the tuning curves of this cell if it had an ATI = 50 ms. Note that now, during head turns in either direction, the cell will fire 50 ms prior to when the animal faces 180 degrees. Our convention takes positive angular head velocity to denote counterclockwise turning. (B) Anticipatory firing of presynaptic HD cells improves speed coding in the postsynaptic LR cell. To quantify strength of speed coding independent of latency between head speed and postsynaptic firing rate, we used the maximum correlation between head speed and postsynaptic firing rate across all time lags. Inset shows a similar relationship, now for the maximum of cross-mutual information. (C) Anticipatory firing of presynaptic HD cells improves the latency between current head speed and postsynaptic firing rate, enabling more temporally precise speed coding.

The online version of this article includes the following figure supplement(s) for figure 9:

**Figure supplement 1.** Analytical calculations independently confirm the improvement of head speed coding with anticipatory firing of head direction inputs.

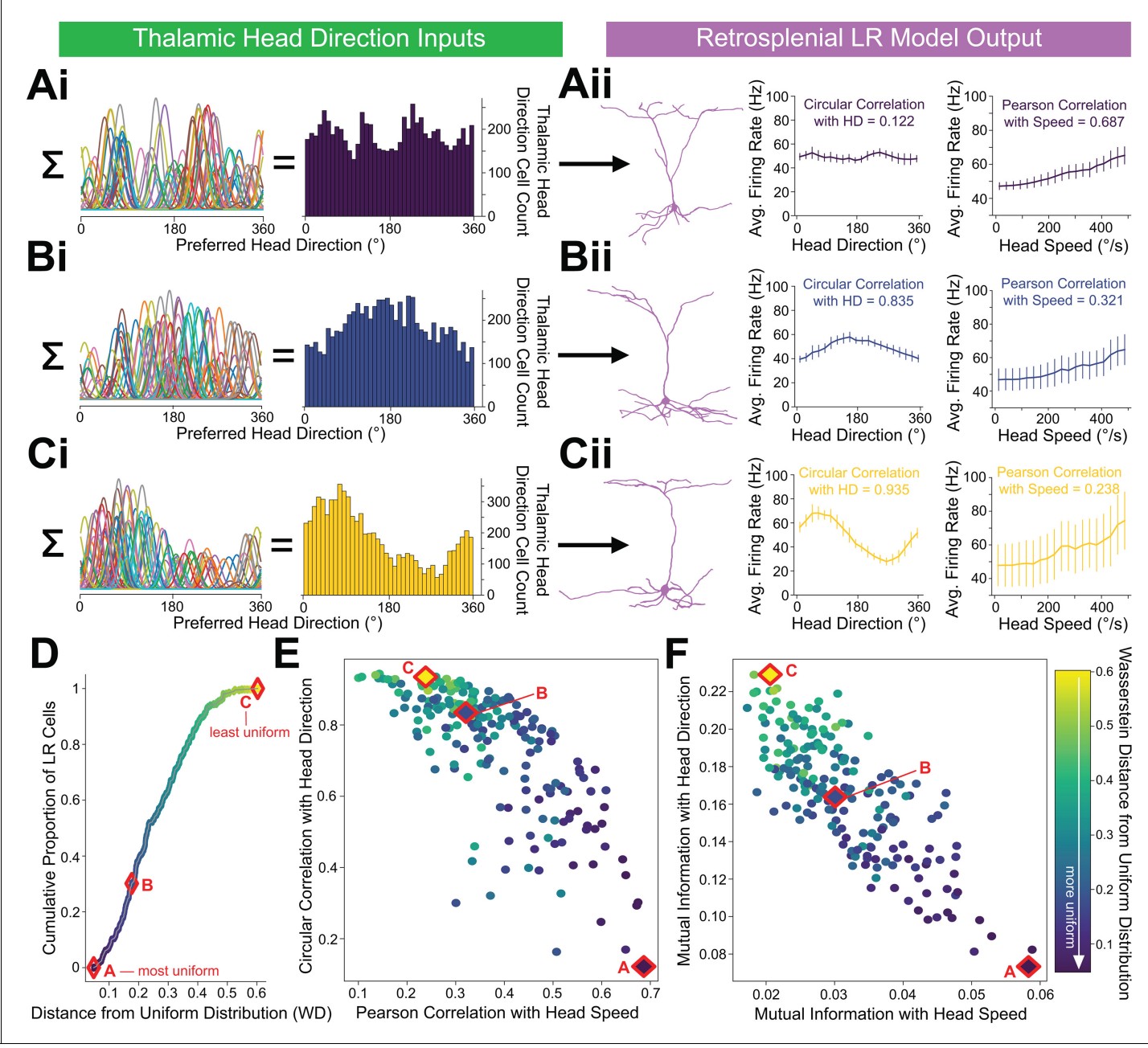

**Figure 10.** LR cells can conjunctively encode both head direction and speed, with a tradeoff. (**A, B, C**) (i) Sample presynaptic preferred HD distributions for three simulated LR cells with distinct heterogeneity among their inputs. A is most uniform, and C is least uniform. Left: schematic depiction of the tuning curves of HD cells synapsing onto the LR cell. Right: calculated histogram of preferred directions of all HD cells synapsing onto the given LR cell. (ii) Response properties of the three LR cells whose preferred direction distributions are depicted in Ai,Bi,Ci. As the preferred direction distribution becomes less uniform (from A to C), directional encoding improves (left graph) while speed coding becomes less precise (right graph). (**D**) For each simulated LR cell, we quantified the heterogeneity of its input HD distribution using the standard metric for this measure (Wasserstein distance from a uniform distribution; WD). Plot shows the cumulative distribution of WD values over the entire simulated LR population. Markers A,B,C correspond to the three example LR cells shown above. (**E**) Scatterplot showing each LR cell's correlation with angular head speed versus its correlation with head direction. Both individually and as a population, LR cells can show conjunctive encoding of head direction and speed, with a clear tradeoff between the two. The color-coded WD is also shown for each cell. (**F**) Same as E but for mutual information.

cannot detect the speed of head rotation from its presynaptic inputs. On the other hand, consider an LR cell that, by chance, gets inputs from a higher proportion of HD cells encoding a direction around 180 degrees. Now, the output of this LR cell is better correlated with direction, acting as a broadly tuned HD cell itself and preferring 180 degrees, but also firing at most other directions. However, this LR cell's encoding of head speed gets worse because it under-samples many of the head directions faced by an animal. Thus, non-uniform HD inputs via depressing thalamocortical synapses represent a simple mechanism by which LR cells can conjunctively encode both head direction and head speed. Overall, our results predict in vivo LR neuronal spike trains that are likely to correlate with both the head direction and rotational speed of an animal, with a tradeoff between direction versus speed coding seen across individual LR cells (*Figure 10E*).

## Discussion

We have identified a circuit that supports parallel processing of information as it first reaches the superficial granular retrosplenial cortex. Layer 3 of RSG contains two distinct pyramidal neuronal subtypes: small, excitable low rheobase (LR) neurons and regular spiking (RS) cells (*Brennan et al., 2020*). Here, we show that LR cells are driven by inputs from the anterior thalamus and dorsal subiculum but are essentially unresponsive to inputs from the claustrum, anterior cingulate, and contralateral retrosplenial cortex. (*Figures 1–6*, *Figure 6—figure supplement 1*). RS cells show precisely the opposite relationship: they are strongly driven by claustral, anterior cingulate, and contralateral retrosplenial inputs, but very weakly activated by anterior thalamic or dorsal subicular inputs (*Figures 1–6*, *Figure 2—figure supplement 1*, *Figure 6—figure supplement 1*). This dichotomy can be explained in large part by the precise overlap of LR versus RS apical dendrites with distinct afferent axons, including clear sublamination of both axons and dendrites within layer 1 (*Figure 6*). Anterior thalamic and dorsal subicular cells show robust directional and spatial modulation (*Barry et al., 2006*; *Olson et al., 2017*; *Stewart et al., 2014*; *Taube, 1998*; *Taube and Bassett, 2003*). Thus, the parallel circuit described here allows LR cells to selectively process this spatially relevant information and makes them ideally suited to support the spatial orientation computations carried out by the RSG.

### LR cells are morphologically and computationally unique

The small, excitable LR cells are unique among pyramidal cells in RSG, and there is no evidence, to our knowledge, for such cells in any other cortical regions (*Brennan et al., 2020*; *Holmgren et al., 2003*; *Jiang et al., 2015*; *Kurotani et al., 2013*; *Wyss et al., 1990*). Indeed, the 'granular' in granular retrosplenial cortex refers to the appearance of these dense small cell bodies in superficial RSG (*Sripanidkulchai and Wyss, 1987*; *Vogt and Peters, 1981*). Thus, LR cells are the defining morphological feature of RSG.

The distinct electrophysiological properties of LR neurons, specifically, play an integral role in the RSG's unique capacity to process anterior thalamic head direction inputs. We have previously reported that the low rheobase and lack of spike frequency adaptation of these neurons are their defining computational features (*Brennan et al., 2020*). For this reason, we refer to them as low-rheobase (LR) neurons. They have also been called late-spiking (LS) neurons (*Kurotani et al., 2013*; *Yousuf et al., 2020*), but we argue this name is not optimal for several reasons. First, many other neurons in superficial RSG are also late spiking (*Brennan et al., 2020*), with L2/3 fast-spiking (FS) interneurons and also some RS neurons exhibiting a substantial delay to first spike, likely due to increased Kv1.1 or 1.2 channel expression in this region compared to other cortical areas (*Kurotani et al., 2013*). Thus, late-spiking is neither a distinctive feature of LR neurons nor excitatory neurons in this region. Second, the name 'late-spiking (LS) neuron' is already the accepted and widely used nomenclature referring to a distinct inhibitory neuronal subtype, the neurogliaform cells that are located throughout the cortical layers in many regions of the brain, including within layer 1 of RSG (*Cruikshank et al., 2012*; *Hestrin and Armstrong, 1996*; *Kawaguchi and Kubota, 1997*; *Overstreet-Wadiche and McBain, 2015*). Instead, the defining computational feature of the small pyramidal cells in L2/3 of the RSG is their low rheobase, and this nomenclature serves to provide an unambiguous name for these hyperexcitable cells (*Brennan et al., 2020*).

## LR cells respond most strongly to inputs from spatially relevant regions

Previous work has reported TC-evoked responses in layer 5 RS neurons at their apical dendrites (*Yamawaki et al., 2019b*), with some variation in the magnitude of responses across cells. Our results reproduce these observations (*Figure 2—figure supplement 1*). Layer 5 is not a homogenous layer, often divided into sublayers 5A and 5B (*Sempere-Ferràndez et al., 2019*; *Sempere-Ferràndez et al., 2018*; *Sigwald et al., 2020*; *Sripanidkulchai and Wyss, 1987*; *Yamawaki et al., 2016a*) and contains a variety of pyramidal cells, including the thin-tufted and thick-tufted neurons who have been shown to exhibit different responses to external inputs (*Sempere-Ferràndez et al., 2018*; *Wyss et al., 1990*). Further work is needed to determine whether any particular subtype of layer five pyramidal neurons may exhibit substantial TC-evoked responses, but our results, reporting significantly larger responses to TC input by LR neurons, suggest LR cells are the predominant spatial information-encoding subtype within RSG.

The dorsal subiculum, which serves to transmit allocentric spatial information such as axis and boundary vector signals (*Lever et al., 2009*; *Derdikman, 2009*; *Olson et al., 2017*; *Simonnet and Brecht, 2019*; *Bicanski and Burgess, 2020*), also precisely targets LR neurons via projections predominantly to layer 3 of RSG (*Figure 5*; *Nitzan et al., 2020*; *Yamawaki et al., 2019a*). These inputs overlap with LR cell bodies, basal dendrites, and apical dendrites in L1b/c and evoke larger excitatory postsynaptic currents in superficial cells compared to layer 5 pyramidal neurons (*Figure 5*; *Nitzan et al., 2020*; *Yamawaki et al., 2019a*). Thus, the distinct morphology and intrinsic properties of LR neurons make them ideally suited to integrate head direction input from the thalamus and spatial inputs from the hippocampal formation via the dorsal subiculum. Indeed, this integration of various types of directional and distance information is often stated as the key computational function of the retrosplenial cortex as a whole (*van Wijngaarden et al., 2020*; *Burgess et al., 2001*; *Byrne et al., 2007*; *Epstein, 2008*; *Ichinohe et al., 2003*; *Maguire, 2001*), further highlighting the defining role that LR cells are likely to serve in the RSG.

## Thalamic inputs to LR cells show short-term depression: implications for angular head velocity and head direction coding in the RSG

We have shown that thalamocortical synapses onto LR cells are depressing (*Figure 7*). Our modeling results demonstrate that such depressing synapses allow LR neurons to compute angular head speed from head direction input, leading to robust encoding of angular speed in the postsynaptic LR firing rate (*Figure 8*). This result is in line with previous studies (*Abbott et al., 1997*; *de la Rocha and Parga, 2008*; *Puccini et al., 2007*) that use synaptic depression to implement neural circuits that perform rate of change computations. We also find that LR encoding of angular head speed is improved by the fact that input HD cells display anticipatory firing (*Figure 9*; *Blair et al., 1997*). This improvement not only allows LR cells to encode current head speed with a shorter time lag (*Figure 9C*), but also strengthens the correlation with head speed regardless of lag (*Figure 9B*). This proposed mechanism by which depressing synapses compute head speed is fully generalizable, and thus may also be of relevance to other brain circuits employing HD-like population codes (*Maunsell and Van Essen, 1983*). Furthermore, we have shown that this thalamocortical synapse can support conjunctive encoding of both head direction and angular head speed by individual LR cells (*Figure 10*).

Our results make specific predictions about the coding properties of LR cells that can be tested with in vivo recordings. First, based on their intrinsic properties (*Figure 1—figure supplement 1*; *Brennan et al., 2020*), LR cells in superficial RSG are likely to show high firing rates and spike widths intermediate to the standard 'broad/narrow' criteria currently used to distinguish putative RS and FS cells in most cortical recordings. Thus, future recordings will need a large sampling of cells in L2/3 of RSG to provide enough data to identify the predicted three distinct clusters of spike shapes (RS/LR/FS). Second, we expect LR cell activity to be correlated with angular head speed with a relatively small lag (*Figures 8* and *9*). Third, we predict that the experimentally observed head velocity tuning curves will be concave up and exhibit approximately quadratic scaling at low rotational speeds, as our analytical work suggests (*Figure 8*; Appendix). Fourth, we anticipate a significant proportion of LR cells will show conjunctive encoding, where both angular head speed and direction are encoded. A key aspect of this conjunctive encoding, as predicted by our simulations (*Figure 10*), is that individual cells will show a tradeoff between directional and speed encoding. As a population, this is

likely to allow LR cells to comprehensively encode conjunctive speed and directional information and provide this information to downstream targets, helping with spatial orientation computations within RSG. This predicted conjunctive encoding will also necessitate that the test of predictions 2 and 3 be carried out in a head-direction delimited manner.

Our modeling work so far has focused on understanding the computational implications of thalamic inputs to LR cells. Future work, beyond the scope of the present study, will be needed to understand how these thalamic inputs interact precisely with dorsal subiculum inputs. Given the diversity of spatial and directional encoding seen in the dorsal subiculum (*Lever et al., 2009*; *Derdikman, 2009*; *Olson et al., 2017*; *Simonnet and Brecht, 2019*; *Bicanski and Burgess, 2020*), computational models will allow for the rigorous undertanding of how each possible type of subicular input (e.g. axis versus boundary vector cells) to LR cells interacts with thalamic head direction inputs. Very recent work on understanding the tuning properties of subicular cells in a projection-specific manner will help to narrow the functional range of possible dorsal subicular inputs to LR cells (*Kitanishi et al., 2021*). We anticipate that one key impact of dorsal subicular inputs to LR cells will be to impose a spatial filter on the angular speed and directional predictions stated above. In particular, the precise organization of layer 1 dendrites and axons may make LR cells most responsive to the near-synchronous arrival of inputs from both thalamic head direction cells (at distal apical dendrites in L1a) and dorsal subicular boundary vector or axis cells (at more proximal apical dendrite locations in lower L1). Similar cooperative dendritic activation patterns are also a hallmark of integration of entorhinal and CA3 inputs by CA1 pyramidal cells, where they help to overcome the effects of strong dendritic inhibition and shape the firing of place cells (*Ahmed and Mehta, 2009*; *Golding and Spruston, 1998*; *Kamondi et al., 1998*; *Takahashi et al., 2012*).

## Downstream targets of LR cells and potential role of RS neurons in retrosplenial spatial computations

LR neurons do not synapse onto neighboring excitatory neurons within layers 2/3 (*Brennan et al., 2020*), and no (0/9) pairs tested in this study between LR and layer 5 RS cells exhibited connectivity (data not shown). Thus, LR cells likely do not provide any substantial input to other LR cells or any RS cells located in layers 2–5 of ipsilateral RSG. Instead, LR neurons send their axons into the corpus callosum and may likely target homotopic regions of contralateral RSG (*Brennan et al., 2020*; *Kurotani et al., 2013*). Indeed, retrograde injections into one RSG hemisphere have been shown to predominantly label small contralateral L2/3 cells (*Sripanidkulchai and Wyss, 1987*; *Van Groen and Wyss, 2003*), most likely corresponding to the LR cells that constitute the majority of cells in L2/3 (*Brennan et al., 2020*). Our supplementary results (*Figure 6—figure supplement 1*) show that inputs from contralateral RSG preferentially and strongly target RS neurons, with almost no recruitment of LR neurons. This is consistent with the observed distribution of afferent axons from contralateral RSG: these axons and terminals avoid L1a and L3 where all LR dendrites are located (*Figure 6—figure supplement 1*). Thus, we propose the following hypotheses for LR downstream targets and possible function: LR neurons in one hemisphere receive unilateral inputs from directional cells in the anterior thalamus (*Figures 1*, *2* and *6*). These inputs are utilized to encode information about both the speed and direction of head rotations and are subsequently transmitted to RS cells in contralateral, but not ipsilateral, RSG (*Figure 6—figure supplement 1*). Thus, LR cells may serve to perform a hemispheric switch of rotational speed and direction information. Given the known bidirectional connectivity between secondary motor cortex and RSG (*Yamawaki et al., 2016a*), RS cells may be able to utilize inputs from contralateral LR cells to perform comparisons of the actual head speed and direction signal with motor efference signals, helping to sharpen the code for spatial orientation in the RSG. However, extensive future investigations of this contralateral circuit are needed to explore these hypotheses.

## Potential role of claustral and anterior cingulate inputs

In contrast to thalamic and DS inputs to RSG, projections from both the claustrum and anterior cingulate cortex avoid laminar overlap with LR apical dendrites, instead projecting to the lower divisions of L1/2 and L5 to target RS apical and basal dendrites (*Figures 3* and *4*, and 6). Both the CLA and ACC are heavily implicated in a range of non-spatial behaviors often described as being 'cognitive' or 'mnemonic' in function (*Botvinick, 2007*; *Brown and Braver, 2005*; *Carter et al., 1998*;

*Devinsky et al., 1995*; *Kim et al., 2016*; *Mohanty et al., 2007*; *Smith et al., 2012*; *Smythies et al., 2012*; *Stevens et al., 2011*), but there is also evidence that these regions exhibit some degree of spatial information processing (*Guterstam et al., 2015*; *Jankowski and O'Mara, 2015*; *Sutherland et al., 1988*; *Whishaw et al., 2001*). Indeed, 7–15% of cells in the claustrum were found to encode place or boundary-related information (*Jankowski and O'Mara, 2015*), though this percentage is relatively low compared to the 60% of anterior thalamic (*Taube, 1998*) and 61% of dorsal subicular (*Kitanishi et al., 2021*) neurons that are known to clearly encode elemental direction, place, and other spatial features. Regardless, retrosplenial RS neurons may be receiving direct spatial inputs from the CLA and ACC in addition to the cognitive and mnemonic information relayed by these two regions. We also speculate that retrosplenial RS neurons may receive directional and spatial information that has already been heavily integrated by LR neurons in the contralateral hemisphere (*Figure 6—figure supplement 1*), as discussed above. This convergence of disparate signals onto RS cells could then serve to support functions that require both spatial and non-spatial integration, such as contextual fear conditioning, a key function attributed to the RSC (*de Sousa et al., 2019*; *Keene and Bucci, 2008a*; *Keene and Bucci, 2008b*; *Kwapis et al., 2015*; *Yamawaki et al., 2019a*).

Indeed, a recent study reported that selective ablation of putative retrosplenial RS cells located near the boundary and within upper layer 5 resulted in contextual fear amnesia (*Sigwald et al., 2020*), indicating that these cells are involved in the fear-related functions of the retrosplenial cortex (*Lukoyanov and Lukoyanova, 2006*; *Yamawaki et al., 2019a*; *Yamawaki et al., 2019b*). The laminar location of these cells corresponds well to the position of the RS cells recorded in our study that we show to preferentially receive CLA and ACC inputs. Both the CLA (*Cho et al., 2017*; *Ipser et al., 2013*; *Vakolyuk et al., 1980*; *Vetere et al., 2017*; *Zingg et al., 2018*) and ACC *Frankland et al., 2004*; *Han et al., 2003*; *Steenland et al., 2012* have been heavily implicated in fear-associated behaviors, suggesting again that CLA and ACC inputs to L3 RS cells help to support the role of these neurons in contextual fear conditioning. Future work that selectively inactivates CLA versus ACC axon terminals in RSC will be necessary to causally confirm the role of each of these synapses in fear conditioning.

## The role of feedforward inhibition

Another important consideration within the circuits examined here is the role of feedforward inhibition. Matrix TC inputs evoke feedforward inhibition to regulate cortical signaling through both disinhibition (*Anastasiades et al., 2021*; *Delevich et al., 2015*) and the disynaptic inhibition of pyramidal neurons (*Cruikshank et al., 2012*; *Yamawaki et al., 2019a*; *Yamawaki et al., 2019b*). Similarly, ClaC inputs invoke feedforward inhibition in several cortical regions, particularly through recruitment of neuropeptide Y (NPY) and, to a lesser degree, FS cells (*Jackson et al., 2018*). Long-range inhibitory signals from CA1 of the hippocampus also target apical dendrites in retrosplenial layer 1a, converging with excitatory anterior thalamic inputs, to precisely regulate these TC inputs and establish a hippocampo-thalamo-retrosplenial network (*Yamawaki et al., 2019b*). This fine-grained laminar overlap strongly suggests that these inhibitory signals may regulate responses to matrix TC input more strongly for LR than layer 5 RS cells and should be investigated further, particularly in the context of integration of spatial information processing. Future work will examine the effect of feedforward inhibition on the subpopulations of retrosplenial principal neurons in response to these converging inputs in order to establish a thorough understanding of the circuitry facilitating retrosplenial information processing, particularly with regard to head direction signals.

## Conclusions

In summary, our results highlight a superficial retrosplenial circuit enabled by the precise sublaminar organization of distinct principal neuronal subtypes and axonal afferents. LR neurons receive directional and spatial inputs from the anterior thalamus and dorsal subiculum. The synaptic dynamics of thalamic inputs to LR cells can give rise to rate coding of angular head speed in LR cells. In contrast, neighboring RS neurons in RSG respond very weakly to these directional inputs, as their apical and basal dendrites distinctly avoid laminar overlap with anterior thalamic and subicular afferents. Instead, RS neurons respond to claustrocortical and anterior cingulate inputs. Determining how these two parallel streams of information are integrated in downstream neurons within both the

ipsilateral and contralateral granular and dysgranular retrosplenial cortices is the next critical step toward a mechanistic understanding of retrosplenial computations.

# Materials and methods

## Key resources table

| Reagent type (species) or resource | Designation | Source or reference | Identifiers | Additional information |
|---|---|---|---|---|
| Strain background (*Mus musculus*) | *Pvalb*^Cre (PV-IRES-Cre) | Jackson Laboratories | Stock# 008069 RRID:IMSR_JAX:008069 | |
| Strain background (*Mus musculus*) | *Camk2α*^Cre | Jackson Laboratories | Stock# 005359 RRID:IMSR_JAX:005359 | |
| Strain background (*Mus musculus*) | *Pvalb*^Cre (PV-IRES-Cre) x Ai14 | Jackson Laboratories | Stock# 008069 Stock# 005359 | Mouselines originally from Jackson Laboratories; in-house cross |
| Strain background (*Mus musculus*) | Tg(*Grp*^Cre)KH288Gsat | Jackson Laboratories | MGI:4367023 RRID:MMRRC_037585-UCD | |
| Strain background (*Mus musculus*) | *Pvalb*^Cre (PV-IRES-Cre) x Ai32 | Jackson Laboratories | Stock# 008069 Stock# 024109 | In-house cross |
| Strain background (*Mus musculus*) | Ai32 | Jackson Laboratories | Stock# 024109 RRID:IMSR_JAX:024109 | |
| Strain background (*Mus musculus*) | Ai14 | Jackson Laboratories | Stock# 007914 RRID:IMSR_JAX:007914 | |
| Peptide, recombinant protein | NeuroTrace 45/455 | Thermo Fisher Scientific | Cat#N21479 | 1:200 |
| Peptide, recombinant protein | Streptavidin, Alexa Fluor 488 Conjugate | Thermo Fisher Scientific | Cat#S11223 | 1:250 |
| Peptide, recombinant protein | Streptavidin, Alexa Fluor 594 Conjugate | Thermo Fisher Scientific | Cat#S11227 | 1:250 |
| Peptide, recombinant protein | Streptavidin, Alexa Fluor 647 Conjugate | Thermo Fisher Scientific | Cat#S21374 | 1:250 |
| Chemical compound, drug | Picrotoxin | Sigma | CAS # 124-87-8, P1675-1Q | 50 µL solution made in house daily from powder |
| Chemical compound, drug | Tetrodotoxin citrate | Alomone Labs | CAS # 18660-81-6 | 1 µL solution made in house daily |
| Chemical compound, drug | 4-Aminopyridine | Alomone Labs | CAS # 504-24-5 | 100 µL solution made in house daily |
| Antibody | Anti-GFP (Chicken polyclonal) | Abcam | ab13970 RRID:AB_300798 | 1:2000 |
| Antibody | Anti-mCherry (Rabbit polyclonal) | Abcam | ab167453 RRID:AB_2571870 | 1:2000 |
| Antibody | Anti-Chicken (Donkey polyclonal), biotin conjugated | Jackson Immuno Research | RRID:AB_2313596 | 1:300 |
| Antibody | Anti-Rabbit (Donkey polyclonal), biotin conjugated | Jackson Immuno Research | RRID:AB_2340593 | 1:300 |
| Software, algorithm | Python 3.7.6 | Python | RRID:SCR_008394 | https://www.python.org/ |
| Software, algorithm | MATLAB R2020a | MATLAB | RRID:SCR_001622 | |

## Animals

All procedures and use of animals were approved by the University of Michigan Institutional Animal Care and Use Committee. The following mouse lines were used for the CRACM experiments: Ai14 (Jackson Laboratories, 007914), Ai32 (Jackson Laboratories, 024109), C57BL6 wildtypes (Charles River, stock #027), *Camk2α*^Cre (Jackson Laboratories, 005359), *Grp*-KH288^Cre (RRID:MMRRC 037585-UCD), *Pvalb*^Cre (PV-IRES-Cre; Jackson Laboratories, 008069), and *Pvalb*^Cre x Ai14 (crossed in

house). A combined total of 70 mice of both sexes between the ages of P41–P184 were used in this study.

## Experimental procedures

### Surgical procedures

Surgical anesthesia was induced via isoflurane inhalation at 4% and then maintained on a surgical anesthetic plane at 2–3% isoflurane. Upon induction, atropine was injected subcutaneously at 0.05 mg/kg. A Physitemp (Clifton, New Jersey, USA) controller monitored and maintained body temperature at 37°C. Ophthalmic ointment was placed on the eyes. The incision site was prepared using 1:40 Nolvasan followed with isopropyl alcohol before being subcutaneously injected with 1% lidocaine. The skull was then leveled, and bregma was identified. Using a digital stereotaxic coordinate system, the following injection target sites were identified: right anterodorsal thalamic nucleus (AP = −0.6 mm, ML = +0.78 mm, DV = −2.5, –3.25 mm), right claustrum (AP = +1.25 mm, ML = +2.6 mm, DV = −3.3 mm), right dorsal subiculum (AP = −3.0 mm, ML = +1.5 mm, DV = −1.5, –1.8 mm), right anterior cingulate cortex (AP = +0.3 mm, ML = +0.3 mm, DV = −0.7, –1.0 mm), and contralateral RSG (see *Figure 6—figure supplement 1*; AP = −2.25 mm, ML = −0.2 mm, DV = −0.7 mm). Burr holes were then drilled through the skull at the identified sites, and dura removed. Micropipettes were lowered under stereotaxic guidance into the target injection site containing the ChR2 viral construct (AAV2-EF1a-DIO-hChR2(h134R)-eYFP for $Grp^{Cre}$ mice and AAV2-hsyn-ChR2(H134R)-eYFP or AAV2-hsyn-ChR2(H134R)-mCherry for all other lines, UNC Gene Therapy Vector Core). For dual injection surgeries, the ChR2 viral construct was injected into the functional target, and a fluorescent tag (AAV2-hsyn-mCherry or AAV2-hsyn-eYFP) was injected into the other region for comparison of various afferent axonal and terminal arborizations within the same animal.

Injections of 0.5 µL total virus volume at each depth were given via a picospritzer at 0.05–0.07 µL/min with a 5-min pause between injecting the more dorsal and ventral DV coordinates, when applicable. After a region was injected, there was a 10-min period before removing the micropipette from the brain. Enrofloxacin was administered at 8.0 mg/kg after injections. Burr holes were sealed with bone wax, and the incision was closed with VetBond with antibiotic ointment placed under skin edges. Isoflurane was tapered down prior to removal. After removal from isoflurane, carprofen was administered at 5 mg/kg. The mice were kept warm through an artificial heat source during the recovery period. Mice then recovered for 2–14 weeks post-injection before being used for slice experiments.

### Slice preparation

Slices were prepared as described previously (*Brennan et al., 2020*). Briefly, mice were deeply anesthetized using isoflurane before decapitation. Brains were removed and placed in a carbogen-saturated ice-cold high-sucrose slicing solution within 30 s of decapitation. Using a Leica 1200VT or Leica 1000S vibratome, 300 µm coronal slices were cut and placed in a high-magnesium artificial cerebrospinal fluid (ACSF) solution at 32°C. After resting in this solution for 20 min, the entire bath was moved to room temperature where the slices rested for the remainder of the experiment.

### Whole-cell recordings

During whole-cell recordings, slices were submerged in a recording chamber with a 2 mL/minute flow of body-temperature ACSF (126 mM NaCl, 1.25 mM NaH2PO4, 26 mM NaHCO3, 3 mM KCl, 10 mM dextrose, 1.20 mM CaCl2, and 1 mM MgSO4). They were visualized using an Olympus BX51WI microscope, Olympus 60x water immersion lens, and Andor Neo sCMOS camera (Oxford Instruments, Abingdon, Oxfordshire, UK). Patch pipettes had a diameter of 2–4 µm and resistances between 2 and 5 MΩ. The potassium gluconate internal solution used in these experiments contained 130 mM K-gluconate, 2 mM NaCl, 4 mM KCl, 10 mM HEPES, 0.2 mM EGTA, 0.3 mM GTP-Tris, 14 mM phosphocreatine-Tris, and 4 mM ATP-Mg and had a pH of 7.25 and osmolarity of 290 mOsm.

Current-clamp recordings were conducted using the Multiclamp 700B and Digidata 1550B (Molecular Devices). Patched neurons were adjusted for series resistances and held at resting potentials around −65 mV. Recordings were not corrected post hoc for liquid junction potential. Resting membrane potential, defined as recorded potential within 30 s of break-in, was recorded, and cells

with resting potentials more depolarized than −50 mV were not included in this study. Intrinsic neuronal properties were calculated using a set of protocols detailed below and measured with either Clampfit or simple custom MATLAB routines (*Figure 1–source code 1*). Pharmacological agents were prepared prior to experiments as outlined by the manufacturer and added to the ACSF. Agents were applied for at least 10 min before conducting experiments, and verification of effect was visualized (no spiking for strong, suprathreshold current injections for TTX-4AP; no feedforward inhibition for picrotoxin).

## Channelrhodopsin-assisted circuit mapping

Channelrhodopsin-assisted circuit mapping (CRACM) experiments (*Yamawaki et al., 2016b*) were conducted under the same rig set-up as described above while using a 5500K white light-emitting diode (LED; Mightex; maximum power of 14.47 mW measured at the slice focal plane). Synaptic responses to optical stimulation of the ChR2-expressing axons were measured from postsynaptic retrosplenial neurons recorded under whole-cell current-clamp conditions.

To examine the effect of afferent axonal input to retrosplenial neurons, two optogenetic stimulation protocols were used. First, a targeted optogenetic stimulation approach was used to examine the distribution of afferent axons and terminal arbors across the cortical layers in which the postsynaptic retrosplenial neurons resided. The main areas of interest included layer 1a (L1a; defined as the top 1/3 of L1 adjacent to the midline), layer 1/2 boundary (L1/2; defined as the border between the sparse L1 and densely-packed L2), the location of the patched cell body (all patched cells were located in layers 3 or 5), upper layer 5 (L5$_{sup}$: defined as 100 µm from the L3/5 boundary), and deep layer 5 (L5$_{dp}$; defined as 200 µm from the L3/5 boundary). For LED-based stimulation over L1a, the objective was centered approximately 30 µm from the pia in L1. For LED-based stimulation over L1/2, the objective was centered over the border between L1 and L2. For LED-based stimulation over the cell body, the objective was centered on the patched cell body at its widest diameter. For L5$_{sup}$, the objective was centered 100 µm into L5, and for L5$_{dp}$, the objective was centered 200 µm into L5. All LED-based stimulations were aligned in the cortical column with the patched cell body. LED intensity was kept constant at each stimulation location and was set at the minimum intensity necessary to evoke the smallest visible response when stimulating at the minimum location (see below). For most cells, this minimal response was between 0.25 and 5 mV, while strongly activated cells had larger responses with the smallest possible LED intensity. The minimum location for ADAV experiments was L1/2, while the minimum location for CLA, ACC, and DS experiments was L1a, corresponding with the lowest projection density of inputs within the superficial layers of RSG. All layer test optogenetic stimulation protocols consisted of a 1 s 10 Hz train of 1 ms LED pulses. For short-term dynamics analyses, an additional optogenetic stimulation protocol was used that consisted of a 1 s 40 Hz train of 1 ms LED pulses.

Second, a current step protocol was used to examine the effect of thalamic input on the postsynaptic retrosplenial neurons' spike trains. In this protocol, a 2 s current step was delivered to the postsynaptic cell that elicited a 10–30 Hz spike train with a 1 ms LED pulse delivered at 500 ms into the current step. The objective for this protocol was centered directly over the cell body.

## Morphological investigations and reconstructions

### Cell filling and visualization

To analyze the morphology of the cells, biocytin (5 mg/ml) was added to the recording solution immediately before recording. Biocytin was allowed to diffuse into the cell for no less than 20 min. Shortly before removing the patch pipette from the neuron, ten current pulses (1-3 nA at 1 Hz) were applied to aid the diffusion process (*Jiang et al., 2015*). After the fill process was complete, the patch pipette was retracted from the cell slowly to allow the membrane resealing. One to four cells were filled per slice, and then the slice was transferred from the recording chamber to 4% PFA for overnight fixation. The next day, slices were washed in PBS and incubated for 24–48 hr in streptavidin conjugated Alexa Fluor (488, 594, or 647) with 0.2% TritonX added to permeabilize the cells. For a subset of cells, fluorescent nissl stain, NeuroTrace, was added to the Alexa Fluor incubation step (at 1:200 dilution). After this incubation, slices were washed in PBS, mounted on slides, and cover slipped using Fluoromount-G.

### Cell reconstruction

Z-stacks of each filled cell were acquired using the Leica SP5 confocal microscope with a dry 40x lens or the Zeiss Axio Image M2 confocal microscope with 20x lens. Reconstructions from z-stacks were performed using NeuroLucida software in user-guided mode or NeuTube software (*Feng et al., 2015*). Additional reconstructions previously generated were also added to the analyses described below (*Brennan et al., 2020*).

### Imaging of slice expression

Representative images of axonal projection to RSG were obtained from mice injected with anterograde AAV2 into ADAV, CLA, ACC, DS, and cRSG. Slices were prepared as for electrophysiology recordings. After use, slices with ADAV, DS, and cRSG expression from AP range −1.7 to −2.06 mm were fixed overnight in 4% PFA and then stained with the fluorescent Nissl stain, NeuroTrace 435–455, to aid laminar demarcation. ACC and CLA slices underwent an additional signal amplification protocol. Those slices were first incubated overnight in PBS containing appropriate normal blocking serums with 0.2% TritonX. On the following day, anti-GFP (for eYFP) or anti-mCherry primary antibodies (Abcam ab13970 and ab167453 respectively, diluted 1:2000) were added and slices were incubated at 4°C on a shaker for up to 24 hr. Next, slices were washed in PBS and incubated on a shaker for 3 hr at room temperature in PBS containing 2% bovine serum albumin (BSA), 0.2% TritonX and biotin-conjugated secondary antibodies (Jackson ImmunoResearch AB_2313596 and AB_2340593, dilution 1:300) then again washed in PBS. Finally, slices were incubated on a shaker in PBS with 0.2% TritonX and streptavidin-conjugated Alexa Fluor (488 for eYFP and 594 for mCherry amplification, dilution 1:300) and NeuroTrace 435–455 (dilution 1:200) for another 3 hr at room temperature, and then washed in PBS. All slices were then mounted on slides using Fluoromount-G and allowed to dry overnight at room temperature. Z-stack images (7 µm at 0.5 µm z-steps) of the slices were obtained the following day using confocal microscope Zeiss Axio Image M2 with 20x dry objective.

## Quantification and statistical analysis

### Neuronal analysis and statistics

Multiple intrinsic neuronal properties were calculated as previously reported (*Brennan et al., 2020*). Briefly, spike threshold, spike amplitude, spike width, spike frequency adaptation ratio, latency to first spike, rheobase, input resistance ($R_{in}$), input capacitance ($C_{in}$), and membrane time constant ($\tau_m$) were measured. Spike amplitude, threshold, and width were calculated from the average of all spikes in a 600 ms current step that elicited a 5 Hz spike train. Amplitude was calculated as the voltage difference from threshold to the peak of the spike, threshold from the peak of the third derivative of membrane potential (*Cruikshank et al., 2012*), and width as the full width at half-maximum spike amplitude. Spike frequency adaptation ratio was calculated from the same 600 ms current step protocol using the first sweep that elicited a 10 Hz spike train using the equation $ISI_{last}/ISI_{first}$. Latency to first spike and rheobase were measured using a 1 s current step protocol that began below threshold and increased by 1–5 pA steps until at least three sweeps post-threshold. Latency to first spike was calculated as the time from the onset of the current step to the peak of the first spike. Rheobase was measured as the minimum current needed to cause at least one spike in the 1 s current step. Lastly, input resistance ($R_{in}$), input capacitance ($C_{in}$), and membrane time constant ($\tau_m$) were all measured from a protocol that delivered a series of small negative current steps resulting in ~4 mV deflection in membrane potential. $R_{in}$ was calculated using Ohm's law, mean voltage change divided by mean current. $\tau_m$ was measured by fitting a single exponential to the average of the initial 60 ms of the negative voltage deflection, ignoring the first 20 ms. Lastly, $C_{in}$ was calculated using the formula $\tau_m = R_{in} \times C_{in}$. Statistical significance of the differences in intrinsic properties between the retrosplenial neuronal subtypes was calculated using the Wilcoxon rank sum test.

### ChR2-assisted circuit mapping analysis and statistics

To quantify the effect of TC and ClaC input across the layers, the amplitude of the resulting postsynaptic responses from each laminar location of LED stimulation (L1a, L1/2, and cell body) were measured with Clampfit and Matlab. The average of the response to the first pulse in the 10 Hz train was calculated for each cell at each LED laminar location. Significant differences between cell types and/

or laminar locations were calculated using the Wilcoxon rank sum test. From the current step protocol, raster plots and PSTHs were plotted for each cell in MATLAB. The percent increase in spike rate was calculated by subtracting the number of spikes in a 100 ms window after the LED pulse from the number of spikes in a 100 ms window before the LED pulse, then dividing this value by the number of spikes in the 100 ms window before the LED pulse. Significant differences between spike count pre- and post-stimulation for both LR and RS cells and LR and RS percent increase in spike rates were calculated using the Wilcoxon rank sum test.

Short-term dynamics were calculated from 10 Hz or 40 Hz LED pulses protocols (see Channelrhodopsin-assisted circuit mapping subsection of the Experimental Procedures section of Methods). To analyze 0.1 Hz short-term dynamics, the first response to the first pulse of each sweep from the 10 Hz protocols was used. Average response amplitudes of each pulse were measured for each cell. Amplitudes were then normalized using the equation stimulus n/stimulus 1, and normalized postsynaptic ratios were plotted. Cells with first pulse responses greater than 0.5 mV but not spiking were included. Significance of resulting short term synaptic depression of LR neuron responses to 10 Hz and 40 Hz thalamic stimulation were calculated using the Wilcoxon rank sum test.

## Morphological analysis and statistics

All z-stack images of axonal projections and their corresponding NeuroTrace channel were first collapsed in ImageJ (using Z project option with max intensity). A section of image spanning layers 1–6 from each image was chosen to serve as a representative sample of expression pattern in that image. In each sample, boundaries of retrosplenial layers 1, 2, 3, and 5 were measured as distance from pia based on NeuroTrace expression, and the information was saved. L1 was distinguished by a low cell density, L2 by a thin, densely packed band of small pyramidal cell bodies, L3 by a lower density (compared to L2) of small pyramidal neurons, L5 as having larger cell bodies relative to L3, and L6 as having smaller cell bodies. Images of fills of the reconstructed cells underwent a similar process (with the exception of LR neurons, which only had layers 1, 2, and 3 boundaries measured, and the average layer width for L5 was applied to them by default). After all images (both cell reconstructions and sample projections into RSG) were measured and processed, they were turned into binary black and white thresholded versions using ImageJ. For the images of reconstructed cells, cell bodies were removed from the black and white images to allow the analysis of dendrites only.

Dendrites and axons were analyzed as a function of cortical depth (*Yamawaki et al., 2014*). White pixels in every image (axonal expression or reconstructed cell) were counted, and their total counts per row were divided by the total numbers of white pixels in the image, resulting in a projection density distribution vector along the layers of RSG. Average layer widths were calculated from the previously acquired measurements and used to set the number of bins per layer to uniformly represent layers. Density plots were then visualized from the adjusted distribution vectors using the following Python modules: pandas, matplotlib, scipy, and Seaborn. For correlation analysis, area from pia to layer 5A, corresponding to the end of RS basal dendrites, was chosen. Spearman's correlation matrices were created using the pandas and scipy Python modules and plotted as a heatmap. Images of axonal projection patterns to prefrontal cortex (PFC) (*Figure 6—figure supplement 2*) were obtained from the Allen Mouse Brain Connectivity Atlas (https://connectivity.brain-map.org; *Oh et al., 2014*).

## Modeling methodology

Our model circuit consists of an RSG LR cell receiving input from an ensemble of $N$ thalamic HD cells, via synapses exhibiting short-term depression.

### Modeling the presynaptic HD population

Throughout, let $\theta_T(t) \in [0, 2\pi]$ denote the head direction at time $t$ ms. Each HD cell has a preferred angle (PA) at which its firing rate is maximized. The firing rate at time $t$ of an HD cell with PA $\theta$ takes the form:

$$r(\theta, t) = f(\theta - \theta_T(t))$$

where $f$ is a fixed function describing the shape of the cell's tuning curve. In all simulations, we utilize a Gaussian tuning curve:

$$f(\theta) = (f_{max} - f_{bg}) \cdot e^{\epsilon(\cos(\theta) - 1)} + f_{bg}$$

Here, $f_{bg}$ is the background firing rate, $f_{max}$ is the maximum firing rate, and $\epsilon$ sets the tuning curve width. We draw these parameters from Beta distributions with statistics in accordance with the ADN HD cell statistics reported in *Taube, 2010*; see *Supplementary file 1* - Table 3 for exact quantities. The spike train of a single HD cell with PA $\theta$ follows an inhomogeneous Poisson process, with intensity given by $r(\theta, t)$ at time $t$. We additionally impose a post-spike refractory period of 4ms.

We draw these parameters from Beta distributions with statistics in accordance with the ADN HD cell statistics reported in *Taube, 2010* (see *Supplementary file 1* - Table 3 for exact quantities). The probability density function of the Beta distribution with parameters $(a, b, \mathrm{loc}, \mathrm{scale})$ is given by $p((x - \mathrm{loc})/\mathrm{scale})$ with

$$p(x) = \frac{\Gamma(a+b) x^{a-1} (1-x)^{b-1}}{\Gamma(a)\Gamma(b)},$$

where $\Gamma$ denotes the standard Gamma function.

Finally, the spike train of a single HD cell with PA $\theta$ follows an inhomogeneous Poisson process, with intensity given by $r(\theta, t)$ at time $t$. We additionally impose a post-spike refractory period of 4ms.

## Anticipatory firing

Thalamic HD cells exhibit an *anticipatory time interval*, or ATI: for each cell, there exists some constant duration Ams such that the cell tends to spike Ams prior to when the animal's heading equals the preferred direction of the cell. We define this notion more precisely following a previous study (*Blair et al., 1997*). Restrict attention to a specific HD cell, with ATI Ams, and let $\theta$ denote its PA when the angular head velocity is zero. The ATI is defined via the property that when angular head velocity equals $\omega$, the PA of this cell shifts from $\theta$ to $\theta - \omega A$. Equivalently, we can think of HD cell PAs as static and instead say anticipatory firing effectively transforms the heading trajectory itself: if $\theta_T(t)$ describes the true angular trajectory, the effective trajectory experienced by our chosen HD cell is:

$$\theta_A(t) = \theta_T(t) + A \cdot \dot{\theta}_T(t)$$

## In vivo head direction recordings

To simulate the angular trajectory $\theta_T(t)$, we employ in vivo mouse head tracking recordings from the data set CRCNS-th1 (*Peyrache et al., 2015*). These in vivo heading recordings were originally sampled at a relatively low rate of approximately 40 Hz; in order to facilitate use of this data at the finer time scale of HD cell spiking, we linearly interpolated the raw data. Angular head velocity was computed using the standard centered approximation to the derivative:

$$\dot{\theta}_T(t_i) \approx \frac{\theta_T(t_{i+1}) - \theta_T(t_{i-1})}{t_{i+1} - t_{i-1}}$$

## Modeling short-term synaptic plasticity

In the Tsodyks-Markram (TM) model (*Tsodyks et al., 1998*) of short-term synaptic plasticity, each synapse has a finite pool from which it releases vesicles into the cleft upon arrival of a presynaptic spike. The state of the synapse at any moment in time is characterized by the triple $(x, u, g)$, specified as:

- $x(t)$: fraction of vesicles remaining in the pool at time $t$.
- $u(t)$: fraction of pool released into cleft upon presynaptic spike arrival at time $t$ (i.e. release probability).
- $g(t)$: synaptic conductance at time $t$.

Here, $t$ is measured in milliseconds, $g$ in umho, and the variables $x$ and $u$ are dimensionless. Depression results from depletion of $x$ due to a spike train, followed by slow recovery of the vesicle pool; facilitation results from increase in $u$ due to a spike train, followed by slow decay of this release probability. Formally, these dynamical variables satisfy the system:

$$\frac{dx}{dt} = \frac{1-x}{\tau_d} - u^+ x^- \delta(t - t_{sp})$$

$$\frac{du}{dt} = \frac{-u}{\tau_f} + U(1-u^-)\delta(t - t_{sp})$$

$$\frac{dg}{dt} = \frac{-g}{\tau_g} + Au^+ x^- \delta(t - t_{sp})$$

Here, $t_{sp}$ denotes an arbitrary presynaptic spike time, and the $\delta$ term specifies a discontinuous modification to the variable upon arrival of a presynaptic spike. We use the notation $f^{\pm}(t) = \lim_{h\to 0^{\pm}} f(t+h)$, and thus $u^+$ and $u^-$ satisfy the relation $u^+ = u^- + U(1-u^-)$. The parameters $\tau_d$, $\tau_f$, and $\tau_g$ set the timescale of depression recovery, facilitation decay, and synaptic conductance decay, respectively. Note that $u^+ \geq U$ always, so $U$ represents the minimum fraction of $x$ contributing to the conductance on each presynaptic spike arrival. Finally, the parameter $A$ is just a tunable synaptic weight (with units of umho).

In the regime of $\tau_d \gg \tau_f$, depression dominates the synaptic dynamics; conversely, in the regime of $\tau_f \gg \tau_d$, facilitation dominates. Accordingly, the limits $\tau_f \to 0$ and $\tau_d \to 0$ describe *depression-only* and *facilitation-only* dynamics, respectively. On the other hand, when $\tau_f$ and $\tau_d$ are on the same order of magnitude, the synapse may exhibit combined depressing-facilitating dynamics.

## Synapse implementation and parameter choices

We utilized an implementation of the Tsodyks-Markram model as a NEURON mechanism, freely available for download from Model DB (https://senselab.med.yale.edu/ModelDB/showmodel.cshtml?model=3815). For all simulations, we use the parameters:

$$\tau_d = 270\text{ms}, \tau_f = 40\text{ms}, U = 0.28$$

in the TM model. We obtained this parameter set by fitting the TM model response to recorded EPSP amplitudes evoked via thalamocortical stimulation of RSG LR cells with a 10 Hz pulse train (*Figures 2* and *7*). This was accomplished by searching parameter space exhaustively for the parameter set minimizing a least-squares loss function. The resultant TM dynamics were moderately depressing, in accordance with the experimental results. Non-depressing synapses were modeled as standard exponential synapses with instantaneous rise time.

## Modeling the postsynaptic LR population

We model the postsynaptic LR cell using the morphologically detailed model originally presented in *Brennan et al., 2020*. This model, implemented in NEURON (*Carnevale and Hines, 2006*), consists of Hodgkin-Huxley conductances representing a fast sodium current, a delayed rectifier potassium current, and a Kv1.1-mediated slow potassium current. We place each HD input → LR synapse at a random location on a distal apical dendrite of the LR cell; varying this synaptic placement does not qualitatively change results (data not shown).

## PA distribution of a postsynaptic cell

Each postsynaptic cell receives input from $N$ presynaptic HD cells, each of which has a unique PA. When studying postsynaptic speed coding, we initially drew the PA of each HD cell uniformly at random from $[0, 2\pi]$. Later on, we examined the effect of introducing nonuniform PA distributions. In this case, we assumed that the PA distribution of each LR cell was continuous as a function of angle, and thus modeled the PA distribution probability densities randomly as resulting from Brownian bridge processes (to ensure $2\pi$ periodicity of the density). We then used inverse transform sampling to draw from the generated distributions.

## Data analysis

To quantify the extent to which postsynaptic activity reflects head turning speed, we utilized two similarity measures $M$: Pearson correlation and mutual information. In fact, we used the time-lagged counterparts of these measures: for a pair of time series $x_t$, $y_t$, their $l$-lagged similarity measure is $M(x_t, y_{t-l})$. Varying $l$, we obtain the cross-correlation and cross-mutual information between the

two variables. Throughout the paper, when referring to an instance of measure $M$ between time series $x_t$, $y_t$, we either report the function $l \mapsto M(x_t, y_{t-l})$ or just the maximum value of the function:

$$\max_l M(x_t, y_{t-l})$$

In order to compute mutual information, we discretized the time series for head turning speed and postsynaptic firing rate into 40 bins each and calculated the mutual information between these discrete proxies.

## Acknowledgements

We thank Scott Cruikshank, Roger Albin, and Clare Weiland for their comments on the manuscript, Vaughn Hetrick for his technical assistance, Michael Roberts for his help with the NeuroLucida software, Victoria Booth for her assistance with the modeling work, and Danny Siu for his assistance with MATLAB-based analyses. This work was supported by lab startup funds from the University of Michigan (OJA), a grant from the Whitehall Foundation (OJA), NIH grant NS121745 (OJA), pilot funds from the NIH/NIA funded Michigan Alzheimer's Disease Research Center 5P30AG05376 (OJA), NIH T-32-NS076401 (EKWB and SPR), and by an NSF graduate research fellowship (EKWB).

## Additional information

### Funding

| Funder | Grant reference number | Author |
| --- | --- | --- |
| University of Michigan | Startup Funds | Omar J Ahmed |
| National Institute of Neurological Disorders and Stroke | NS121745 | Omar J Ahmed |
| Whitehall Foundation | Research Grant | Omar J Ahmed |
| National Institute on Aging | 5P30AG053760 | Omar J Ahmed |
| National Science Foundation | Graduate Research Fellowship | Ellen KW Brennan |
| National Institute of Neurological Disorders and Stroke | T-32-NS076401 | Ellen KW Brennan Sharena P Rice |

The funders had no role in study design, data collection and interpretation, or the decision to submit the work for publication.

### Author contributions

Ellen KW Brennan, Sameer Kailasa, Conceptualization, Software, Formal analysis, Investigation, Visualization, Methodology, Writing - original draft, Writing - review and editing; Izabela Jedrasiak-Cape, Conceptualization, Software, Formal analysis, Investigation, Visualization, Methodology, Writing - review and editing; Sharena P Rice, Investigation, Writing - review and editing; Shyam Kumar Sudhakar, Software, Writing - review and editing; Omar J Ahmed, Conceptualization, Software, Supervision, Funding acquisition, Visualization, Methodology, Writing - original draft, Project administration, Writing - review and editing

### Author ORCIDs

Sameer Kailasa https://orcid.org/0000-0002-9816-6345
Omar J Ahmed https://orcid.org/0000-0003-3300-7658

### Ethics

Animal experimentation: All procedures and use of animals were approved by the University of Michigan Institutional Animal Care and Use Committee (Protocol #9572). This study was performed in strict adherence with guidelines by the Guide for the Care and Use of Laboratory Animals and the Institutional Animal Care and Use Committee. Surgical procedures were performed under isoflurane

anesthesia, and analgesics and other care was used to minimize animal discomfort. Slicing protocols also used deep isoflurane anesthesia to eliminate any pain or discomfort felt by the animal.

## Decision letter and Author response

Decision letter https://doi.org/10.7554/eLife.62207.sa1
Author response https://doi.org/10.7554/eLife.62207.sa2

## Additional files

### Supplementary files

• Supplementary file 1. Supplemental Tables. contains Table 1, Table 2, and Table 3. Table 1 presents mean ± SEM values of intrinsic properties of LR, L3 RS, and L5 RS cells. Table 2 presents the exact p-values evaluating differences between the three cell types for each intrinsic property examined. Source data files for these intrinsic properties are found in *Figure 1—source data 1*. Table 3 presents exact parameter distributions used for the model (see 'Modeling the Presynaptic HD Population').

• Transparent reporting form

### Data availability

The LR neuron model utilized here is available on ModelDB (https://modeldb.yale.edu/260192), and the implementation of the Tsodyks-Markram model utilized here is also available from ModelDB (https://senselab.med.yale.edu/ModelDB/showmodel.cshtml?model=3815). Reconstructions will be available at http://neuromorpho.org/KeywordSearch.jsp and can be found by searching for the "Ahmed" archive. Source experimental data for figures 1-5, 7, Figure 1-Figure supplement 1, and Supplementary File 1-Table 1 are provided.

The following previously published datasets were used:

| Author(s) | Year | Dataset title | Dataset URL | Database and Identifier |
|---|---|---|---|---|
| Brennan EKW, Sudhakar SK, Jedrasiak-Cape I, John TT, Ahmed OJ | 2020 | Two populations of excitatory neurons in the superficial retrosplenial cortex (Brennan et al 2020) | http://modeldb.yale.edu/260192 | ModelDB, 260192 |
| Tsodyks M, Pawelzik K, Markram H | 2000 | Synaptic plasticity: pyramid->pyr and pyr->interneuron (Tsodyks et al 1998) | http://modeldb.yale.edu/3815 | ModelDB, 3815 |

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

# Appendix 1

## Mean-field model analysis

In order to better understand the observed speed tuning effects, we study an analytically tractable mean-field model of the simulation circuit. We show for this model that the aggregate postsynaptic activity does indeed reflect head turning speed – in fact, at low speeds it is proportional to the squared head speed. Moreover, we show that anticipatory firing compensates for lag introduced by the depression recovery timescale, enabling sharper and lower latency speed encoding. Finally, we briefly comment on model predictions when the HD synaptic input exhibits facilitating or combined facilitating-depressing dynamics.

### Model description

Let $\theta_T(t)$ denote the true head direction at time $t$. For our mean-field analysis, we model all HD cells sharing preferred angle (PA) $\theta$ as a lumped subpopulation, whose firing rate at time $t$ is given by $r(\theta,t) = f(\theta - \theta_T(t))$ for the tuning curve function

$$f(\theta) = \left(f_{max} - f_{bg}\right) \cdot 1_{(-\epsilon,\epsilon)}(\theta) + f_{bg} \tag{1}$$

Here $1_I$ denotes the indicator function of the interval $I$. Note that this is not the same tuning function as utilized in simulations; here, the subpopulation of HD cells with PA $\theta$ is active precisely at times $t$ when the current head direction $\theta_T(t)$ is within $\epsilon$ of $\theta$. This simplification, in which HD cell tuning curves are discontinuous step functions rather than continuous Gaussians, gives qualitatively analogous results to the Gaussian case (data not shown). In the mean-field model, we further assume the background firing rate $f_{bg}$, maximum firing rate $f_{max}$, and tuning curve width $\epsilon$ are constants independent of $\theta$.

The synapses follow simplified Tsodyks-Markram (TM) dynamics (**Tsodyks et al., 1998**), in which the dynamical quantities represent population-averaged values of the standard TM synaptic variables (see Materials and methods for details of the standard TM model) over all synapses corresponding to HD cells having a particular $\theta$. In what follows, we explicitly model synaptic depression only, although similar equations describe the full TM model with combined depression and facilitation. Thus, we are interested in the quantity $x(\theta,t)$, representing the average value of the depression variable $x$ for all synapses corresponding to HD cells with PA $\theta$, at time $t$. The dynamics of $x$ are given by the equation:

$$\frac{dx(\theta,t)}{dt} = \frac{1 - x(\theta,t)}{\tau} - Ux(\theta,t)r(\theta,t) \tag{2}$$

Here $\tau$ denotes the time constant of recovery from depression, and $U$ represents the synaptic release probability. When $x(\theta,t) \approx 0$, the HD cells with PA $\theta$ are very depressed; when the firing rate $r(\theta,t)$ lets up, $x$ recovers to its steady-state value of 1. Similarly, $g(\theta,t)$ denotes the average conductance of all synapses corresponding to HD cells with PA $\theta$, at time $t$. We consider the limit in which the timescale of synaptic decay is much shorter than the depression recovery timescale $\tau$, and thus we can make the approximation $g(\theta,t) = x(\theta,t) \cdot r(\theta,t)$.

Now, let $p(\theta)$ denote the density function of the presynaptic PA distribution. We are interested in dynamics of the $\theta$-averaged quantities

$$\bar{x}(t) = \int_0^{2\pi} x(\theta,t)p(\theta)d\theta$$

$$\bar{g}(t) = \int_0^{2\pi} g(\theta,t)p(\theta)d\theta \tag{3}$$

In particular, $\bar{g}$ is proportional to the sum of $g$ over all $\theta$, i.e. the total conductance of the postsynaptic cell, and thus reflects the activity level of the postsynaptic cell.

## Equation for dynamics of $\bar{g}$

In this section, we derive an explicit equation for the dynamics of $\bar{g}$ in the limit of low background firing ($f_{bg} \to 0$). Crucially, the choice of step function tuning enables us to close the relevant system of ODEs to obtain low-dimensional dynamics. Although we will not study this case in any further depth, we note that when $f_{bg}$ is nonneglible, the quantities $\bar{x}$ and $\bar{g}$ form a coupled planar system.

### Proposition

In the limit of low background firing ($f_{bg} \to 0$), the mean synaptic conductance $\bar{g}$ satisfies the equation

$$\frac{d\bar{g}}{dt} = \frac{b_g(t) - \bar{g}}{\tau_g} + f_{max}\dot{\theta}_T(t)\alpha(t) \tag{4}$$

Here we have

$$\alpha(t) = x(\theta_T(t) + \epsilon, t)p(\theta_T(t) + \epsilon) - x(\theta_T(t) - \epsilon, t)p(\theta_T(t) - \epsilon) \tag{5}$$

$$b_g(t) = \frac{\bar{r}(t)}{1 + \tau U f_{max}} \tag{6}$$

and $\tau_g = \frac{\tau}{1+\tau U f_{max}}$, with $\bar{r}$ defined as the average firing rate of a presynaptic HD cell at time $t$, i.e.

$$\bar{r} = \int_0^{2\pi} r(\theta, t)p(\theta)d\theta = (f_{max} - f_{bg})\int_{\theta_T(t) - \epsilon}^{\theta_T(t) + \epsilon} p(\theta)d\theta + f_{bg}$$

### Proof

From the defining equation (**Equation 2**) for $x$, we see that $\bar{x}$ satisfies

$$\frac{d\bar{x}}{dt} = \frac{1 - \bar{x}}{\tau} - U\bar{g}$$

We have for fixed $\theta$ that

$$\frac{dg}{dt} = \frac{dx}{dt}r + \frac{dr}{dt}x = \frac{r-g}{\tau} - Uxr^2 + x\frac{dr}{dt}$$

For our specific choice of step function $r$, we have $r^2 = (f_{max} + f_{bg})r - f_{max}f_{bg}$. Setting $C = f_{max} + f_{bg}$ and $D = f_{max}f_{bg}$, we thus have

$$\frac{dg}{dt} = \frac{r-g}{\tau} - UCxr + UDx + x\frac{dr}{dt} = \frac{r-g}{\tau} - UCg + UDx + x\frac{dr}{dt}$$

Now averaging over $\theta$ gives

$$\frac{d\bar{g}}{dt} = \frac{\bar{r} - \bar{g}}{\tau} - UC\bar{g} + UD\bar{x} + \int_0^{2\pi} xp\frac{dr}{dt}d\theta = \frac{\bar{r}/(1 + \tau UC) - \bar{g}}{\tau/(1 + \tau UC)} + UD\bar{x} + \int_0^{2\pi} xp\frac{dr}{dt}d\theta$$

Setting $b_g = \bar{r}/(1 + \tau UC)$ and $\tau_g = \tau/(1 + \tau UC)$, we can rewrite this as

$$\frac{d\bar{g}}{dt} = \frac{b_g - \bar{g}}{\tau_g} + UD\bar{x} + \int_0^{2\pi} xp\frac{dr}{dt}d\theta$$

$$\frac{dr}{dt} = -\dot{\theta}\frac{dr}{d\theta} = -E\dot{\theta}(\delta(\theta - \theta_T(t) + \epsilon) - \delta(\theta - \theta_T(t) - \epsilon))$$

where $\delta$ denotes the Dirac delta function and $E = f_{max} - f_{bg}$. It follows that

$$\int_0^{2\pi} xp\frac{dr}{dt}\,d\theta = E\dot{\theta}(x(\theta_T(t)+\epsilon,t)p(\theta_T(t)+\epsilon) - x(\theta_T(t)-\epsilon,t)p(\theta_T(t)-\epsilon)) = E\dot{\theta}\alpha$$

Finally setting $f_{bg} = 0$ gives the proposition. ∎

## Analyzing Contributions To $\bar{g}$

In the $f_{bg} \to 0$ limit, the mean conductance $\bar{g}$ is driven by fluctuations in $\dot{\theta}\alpha$, between which it relaxes to the value $b_g$ with time constant $\tau_g$. Accordingly, to understand the behavior of $\bar{g}$ we should study the behavior of $b_g$ and $\alpha$. For all of what follows, we assume $f_{bg} = 0$.

### Baseline activity: $b_g$

When $\dot{\theta}_T = 0$, i.e. the true head direction is not changing, we have $\bar{g} = b_g$ following a transient. Thus, we can interpret $b_g$ as the baseline activity level of the postsynaptic cell. In the case $p(\theta) = 1/(2\pi)$, i.e. that of a uniform presynaptic PA distribution, the baseline activity $b_g$ is constant in time, since:

$$b_g \propto \bar{r} \propto \int_{\theta_T(t)-\epsilon}^{\theta_T(t)+\epsilon} p(\theta)d\theta = \frac{\epsilon}{\pi}$$

On the other hand, if $p(\theta)$ is nonuniform, then the integral defining $\bar{r}$ is not constant in time, taking larger values when $\theta_T(t)$ is at angles around which the PA distribution $p$ has more mass. Since $b_g \propto \bar{r}$, $b_g$ exhibits the same effect, thereby explaining the observed modulation of baseline firing rate by head direction.

### Contribution of depression: $\alpha$

We now assume in all of what follows that the PA distribution is uniform, i.e. $p(\theta) = 1/(2\pi)$. In this case, the function $\alpha$ reduces to

$$\alpha(t) = \frac{1}{2\pi}(x(\theta_T(t)+\epsilon,t) - x(\theta_T(t)-\epsilon,t))$$

We can interpret $\alpha$ as the contribution of synaptic depression to $\bar{g}$, since it is the only term in the equation for $\bar{g}$ that incorporates any effect of the depression variable $x$. To understand the behavior of $\alpha$, we differentiate to obtain

$$\frac{d\alpha}{dt} = \dot{\theta}_T\beta - \frac{\alpha}{\tau_\ell} \tag{7}$$

where $\beta(t) = x_\theta(\theta_T(t)+\epsilon,t) - x_\theta(\theta_T(t)-\epsilon,t)$ and

$$\tau_\ell = \frac{\tau}{1 + \tau U f(\epsilon)} \tag{8}$$

Note that for our step function tuning, $f(\epsilon)$ is not actually defined. However, we can treat $f$ as if it is continuous at $\epsilon$. One way to justify this is to imagine $f$ has been replaced with a continuous function $h$ that is close to $f$ in the sense that

$$\max_{x\in[0,2\pi]} |f(x) - h(x)|$$

is small; it is always possible to make this quantity arbitrarily small with a suitable choice of continuous $h$.

The equation above further reduces the problem of understanding $\alpha$ to that of understanding $\beta$, which distorts the velocity information $\dot{\theta}_T$ appearing in (*Equation 7*). To proceed, we make the approximation $\beta(t) \approx M$, where $M>0$ is the time-average of $\beta$. This approximation will be valid if $\beta$ is typically positive and not too variable about its mean. Indeed, at each moment $t$ in time, we have a depression profile $\theta \mapsto x(\theta,t)$ describing, for PA $\theta \in$, the average depression of synapses from the HD subpopulation having PA $\theta$. The graph of this profile has a well-like shape, centered close to $\theta_T(t)$. At low speeds, the slope $x_\theta(\theta_T(t)+\epsilon,t)$ will be positive, and the slope $x_\theta(\theta_T(t)-\epsilon,t)$ will be negative;

it follows that the difference $\beta$ is itself typically positive, becoming smaller only during high speed head turns. For now, we assume $\beta$ also has low variability, and that therefore the approximation $\beta \approx M$ holds; later on, we will discuss the validity of this assumption. Continuing with the approximation, we have the simplified equation

$$\frac{d\alpha}{dt} = M\dot{\theta}_T - \frac{\alpha}{\tau_\ell} \tag{9}$$

It follows that $\alpha$ tracks the variable $M\dot{\theta}_T$, i.e, we have the approximate relation $\alpha \propto \dot{\theta}_T$. Since $\bar{g}$ is driven by the product $\dot{\theta}_T \alpha$, it follows that approximately $\dot{\theta}_T \alpha \propto \dot{\theta}_T^2$. Thus, when the approximation for $\beta$ is valid, we conclude the postsynaptic conductance $\bar{g}$ is proportional to the square of angular head speed.

However, these proportionality relations are only approximate, and are degraded by the integration times $\tau_\ell$ in (*Equation 9*) and $\tau_g$ in (*Equation 4*). In the next section, we will see that anticipatory firing compensates for these lags and sharpens speed coding.

## Anticipatory firing improves speed coding

As mentioned above, the lags $\tau_l$ and $\tau_g$ degrade the approximate relation $\bar{g} \propto \dot{\theta}^2$. Indeed, as shown in *Figure 10*, at ATI = 0 the postsynaptic activity is not especially informative about head speed, relative to the ATI = 50 ms and ATI = 100 ms cases. Intuitively, a positive ATI compensates for these lags; to make this statement precise, we note the following lemma, for which we omit the straight forward proof.

### Lemma

Let $F$ be a differentiable function with $F(0) = 0$. If $C = \tau$, then the solution $y$ to the initial value problem

$$\frac{dy}{dt} = F(t) + CF'(t) - \frac{y}{\tau}$$

$$y(0) = 0$$

has the property that $y(t) = \tau F(t)$ following a transient (more precisely, $y - \tau F \to 0$ exponentially fast as $t \to \infty$)

—

Recall that introducing an ATI of A ms is equivalent to replacing the true angular trajectory $\theta_T(t)$ with the effective trajectory $\theta_A(t) = \theta_T + A\dot{\theta}$. Applying the lemma to (*Equation 9*) with $C = A = \tau_\ell$ and $F = \theta_T$, we see that when the ATI equals $\tau_\ell$ ms, the relation $\alpha \propto \dot{\theta}_T$ becomes *exact* (with a proportionality constant $C\tau_\ell$). Moreover, in this case, the equation for $\bar{g}$ simplifies to

$$\frac{d\bar{g}}{dt} = \frac{b_g - \bar{g}}{\tau_g} + f_{max}\dot{\theta}_A \alpha = \frac{b_g - \bar{g}}{\tau_g} + f_{max}C\tau_\ell(\dot{\theta}_T + A\ddot{\theta}_T) \cdot \dot{\theta}_T$$

Since we have assumed $p(\theta)$ is uniform, and hence $b_g$ is constant in time, we can apply the lemma again taking $F(t) = (b_g/\tau_g) + f_{max}C\tau_\ell\dot{\theta}_T^2$. If we assume $C = A = 2\tau_g$, then there is the exact equality

$$\bar{g} = b_g + f_{max}C\tau_\ell\tau_g\dot{\theta}_T^2$$

Thus, the condition for exact speed coding is

$$A = \tau_\ell = 2\tau_g$$

Finally, we note that in (*Equation 8*) the most parsimonious assumption to make is that $f(\epsilon) \approx f_{max}/2$; this amounts to assuming we have replaced the step function tuning $f$ with a continuous function $\tilde{f}$ whose steepest portions are centered at $\epsilon$ and $-\epsilon$. It follows that indeed $\tau_\ell \approx 2\tau_g$, and therefore, it is possible to be in the regime of exact speed coding. This analysis shows that the

ATI optimizing speed coding in our model is $2\tau_g$. For the parameter set used in simulations ($\tau$ = 270ms, U = 0.28ms, $f_{max}$ = 0.07 spikes/ms), we find that

$$\tau_g = \frac{\tau}{1 + \tau U f_{max}} \approx 43\text{ms}$$

Accordingly, the optimal ATI is $2\tau_g \approx 86\text{ms}$. Indeed, simulations utilizing the above parameter set and step-function tuning curves exhibit optimal speed coding at an ATI of precisely 85 ms, thereby validating the analysis (*Figure 8—figure supplement 1*).

## Scaling of postsynaptic activity with speed

As mentioned earlier, validity of the approximation $\beta(t) \approx M$ depends on the variability of $\beta$ about its mean. When this approximation is valid, we have the scaling $\overline{g} \propto \dot{\theta}_T^2$. Before describing the empirical validity of this approximation, we first note that even when $\beta$ is fairly variable, the function $\alpha$ remains strongly correlated with $\dot{\theta}_T$, and consequently the analysis of ATI in the previous section remains approximately valid. Indeed, in the case of step-function tuning, we find that $\beta$ is actually fairly variable (CV = 0.6), but postsynaptic activity is still very strongly correlated with speed at the optimal ATI computed by the formula above (*Figure 9—figure supplement 1*). However, the distortion by $\beta$ of the velocity information in $\alpha$ results in a speed-postsynaptic activity relation whose best-fit power law exponent is approximately 1.5-1.7, instead of the predicted value of 2. Thus, for step function tuning, the quadratic scaling does not hold.

On the other hand, for Gaussian tuning curves, the approximation $\beta(t) \approx M$ is valid. Although the above derivations have assumed step-function tuning, we may heuristically imagine that the equations are approximately valid even with Gaussian tuning, for some suitable choice of tuning curve width $\epsilon$. Proceeding as such, we find that with Gaussian tuning, $\beta$ is relatively invariable about its mean (CV = 0.2), especially for low head speeds. We note that the tuning curve shape directly affects the shape of the depression profile $\theta \mapsto x(\theta, t)$, and consequently affects the variability of $\beta$. This suggests that for low speeds and Gaussian tuning, the scaling $\overline{g} \propto \dot{\theta}_T^2$ will be accurate. Mean-field simulation results do yield that for low speeds, the best-fit power law exponent is approximately 1.9-2.1.

This raises the question of whether this scaling is observable in data from simulations of our spiking model, since spiking simulations utilize the Gaussian tuning, which is more accurate to the experimentally observed shape of HD cell tuning curves. We fit results from the spiking model (postsynaptic firing rates binned by head speed) to a power law when including data points in the speed-firing rate curve only up to $y$ deg/s, for varying values of $y$ (*Figure 8—figure supplement 2*). We found that best-fit residuals were lowest for low speeds and that the best-fit exponents plateaued in a neighborhood of 2 before falling off as the considered range of speeds widened, suggesting that the approximate quadratic scaling is potentially visible in experimental data in this circuit and others utilizing the same synaptic-depression-based mechanism for computing rate of change.

