## [Decision Letter]

**Acceptance summary:**

The authors have offered strong and elegant evidence of a circuit that supports parallel processing of information in the granular retrosplenial cortex (RSG). Notably, this parallel processing identified by the authors provides a synaptic and circuit mechanism that likely supports the role of the RSG in spatial behaviors.

**Decision letter after peer review:**

Thank you for submitting your article "Thalamus and claustrum control parallel layer 1 circuits in retrosplenial cortex" for consideration by *eLife*. Your article has been reviewed by 3 peer reviewers, and the evaluation has been overseen by a Reviewing Editor and Laura Colgin as the Senior Editor. The reviewers have opted to remain anonymous.

The reviewers have discussed the reviews with one another and the Reviewing Editor has drafted this decision to help you prepare a revised submission.

All three reviewers consider that the study addresses important questions in regard to the organization of the neuronal circuits of the granular retrosplenial cortex (RSG). Specifically, the observation that afferent input from the claustrum and the anterior thalamus synapse onto different classes of RSG neurons, is novel and well supported by the present data. However, the reviewers raised a series of concerns that must be addressed before we can reach a decision on the manuscript.

While we consider that the points raised by the reviewers must be addressed in full, particular attention should be placed on addressing how the RSG can perform independent spatial and fear computations. Indeed, whereas the connectivity data presented here is suggestive of functional segregation within the proposed circuits, the behavioral and computational implications of these findings are at present highly speculative. The reviewers suggest that a connection between the present findings and circuit computation could be achieved via a model or further experimentation as described in detailed below.

*Reviewer #1:*

Brennan et al. present data concerning inputs to layer 1 of retrosplenial cortex arising from the anterior thalamus and claustrum. They provide strong evidence for non-overlap of inputs to retrosplenial cortex from these two structures that segregate both in terms of the specific layer 1 sub-regions and layers III/IV sub-regions that are targeted as well as two different groups of pyramidal cell types that are identified electrophysiologically. The two types are "low-rheobase" (LR) neurons and regular-spiking (RS) neurons. The anatomical and electrophysiological claims are well-evidenced and the data is presented at a time in which there is great interest in retrosplenial cortex and its lamina-specific inputs. There is a nice complement to work concerning inputs from subiculum and hippocampus. The authors also examine to some extent the differential tuning properties of LR and RS neurons in combined modeling/electrophysiology experiments.

I find the work interesting, but some of the modeling work seems a bit derivative of work done in a prior publication by the same group (Cell Reports). In addition, the manuscript suffers from an over-interpretation based on the firing properties of anterior thalamic and claustrum neurons. The attribution of claustrum inputs as "higher-order" and "fear-related" goes much too far. Finally, the work lacks an experimental consideration of how the two groups of layer III pyramidal cells are integrated within retrosplenial cortex. For this reason, the emphasis on "parallel circuits" may be misleading. It is not clear if (at the extreme), there are independent circuits in retrosplenial cortex that happen to co-localize to the same region (perhaps to gain input from a third common source) or if the segregation at the level of layer III is followed by integration (for example, by layer V neurons). The combination of anatomy, modeling and electrophysiology has merit, but some of the findings have precedent and, most importantly, the interpretation of the data with respect to the retrosplenial cortex as a whole needs improvement.

1. While it is mostly reasonable to speak of anterior thalamic inputs as reflecting head direction inputs (with high input rates), the claustrum inputs cannot be described this way. Unfortunately, there is very little data concerning the response properties of claustrum inputs (though data from O'Mara lends some suggestion of spatial tuning). In addition, a brain region with neurons responding in fear/avoidance based learning scenarios (or for which lesions impair such learning) is all too common. Neurons throughout the brain respond in some way in such conditions as they often do to reward. The authors should refrain from tagging the claustrum inputs in particular with psychology jargon. I realize that this leaves the meaning of the claustrum input as a bit of a mystery, but perhaps other more neuroanatomy or neurophysiology based arguments for the special properties of a claustrum input can be advanced. The authors might also find it much easier to advance arguments of function by putting more focus into the anterior cingulate inputs (cingulate responses are much better defined at this point).

2. Much attention is paid to the layer 1 inputs, but the two input streams diverge equally with respect to layer 3-5. More discussion of the importance of this is desirable and might begin with why each input stream has two very different concentrations of lamina-specific input. What neurons and dendrites are targeted by the layer 3-5 inputs and how does this relate to the layer 1 inputs. In general, the manuscript does not go far enough to incorporate details of circuitry between neurons of retrosplenial cortex. The absence of this leaves the manuscript lacking some originality in terms of data presented and conceptual issues.

3. The data presented in figure 4 is compelling with respect to latency to response (5ms is a long time in the brain), but I encourage the authors to spend more time working through the description in results. As written, I think many readers will need to re-read multiple times to understand what is going on (and probably won't re-read). Please make this section more accessible to the broader set of neuroscientists without special expertise in this kind of work.

4. Can the authors provide greater context by speaking to the issue of axonal targets of retrosplenial LR and RS neurons?

*Reviewer #2:*

In this manuscript, Brennan et al. expand upon their previous observation of two different pyramidal neuron subtypes in RSG. They had recently shown that RS and LR neurons in superficial RSG exhibit distinct intrinsic excitability profiles and axo-dendritic arborization. Here, they provide evidence that this is paralleled by specific connectivity of long-range inputs and corresponding synaptic dynamics in the two cell types. The authors frame their work through the question of how RSG subserves both fear and spatial coding, concluding that these streams of information are processed by separate and largely independent circuits in RSG built around RS and LR neurons.

This is an exciting topic that is potentially a good fit for *eLife*. How differential connectivity interacts with synaptic and/or intrinsic properties of distinct cell types to shape the computations that contribute to behavioral processes is a pressing problem in the field. And it is highly challenging to address. The specific observation at the core of the manuscript, that thalamic inputs preferentially target LR neurons while claustrum inputs preferentially target RS neurons, is quite interesting. However, the manuscript itself is preliminary and poorly organized, with a number of loose ends and an unclear impact. Substantial revision and expansion are necessary to do justice to the importance of the initial differential connectivity result.

1. The authors should finish the incomplete experiments shown in the supplement, which look potentially highly complementary to the data presented in the main figures. Specifically, the authors should complete the subiculum and ACC input anatomy, connectivity, and synaptic dynamics to LR and RS neurons. It is currently a serious missed opportunity. The resultant data will increase the impact of the paper, and it will be particularly useful for model building.

2. The manuscript should be reorganized to present the comprehensive anatomy, connectivity, and synaptic dynamics of AT, CLA, ACC, and SUB in a principled way. Right now, the figures are all over the place, making comparisons challenging. For example, why is there no parallel figure to Figure 3 for CLA inputs? And why are some of the experiments repeated in L5 RS neurons? Is there a parallel system of RS and LR neurons in L5? Have the authors just not found the LR analogues in L5? I am not sure if the L5 data even adds anything – right now it seems like more of a distraction. Maybe it's best to stick to just L2/3.

3. The authors assert (in the abstract, intro, and discussion) that they have identified and modeled how RSG can perform independent spatial and fear computations. They did not test this at all and should not push this agenda without specific, concrete experiments and data that support this direction. What the authors have shown is differential connectivity and synaptic dynamics for two inputs to two cell types. That is basically it. The biophysical model in Figure 4 really has nothing to do with spatial or HD coding. And Figure 7 is not a model at all, but a rudimentary cartoon schematic. In order for this manuscript to have broad impact in the field, the authors should perform a serious modeling effort that integrates what is known about the firing properties of the different neurons that output to RSG with the connectivity and synaptic dynamics they have discovered at RS and LR neurons to test how this system works in vivo. Receptive field properties of RSG neurons have previously been reported by multiple groups (Nitz, Jeffreys, Smith, Bonin, etc): the authors should develop a model that captures some of these in vivo properties and makes non-trivial, testable, and quantitative predictions about, for example, how parallel LR/RS circuitry is beneficial for some aspect of encoding.

*Reviewer #3:*

The authors distinguish between two previously characterized types of principal pyramidal neurons in Layer 3 of the granular retrosplenial cortex (RSG): low-rheobase (LR) and regular-spiking (RS) neurons, and identify them by their unique intrinsic properties using patch clamp.

Viral injections into the anterodorsal and anteroventral thalamic nuclei (AD/AV) resulted in ChR2-eYFP being expressed in thalamocortical (TC) axons and terminals in layer 1a (L1a; the most superficial part of layer 1) and in layer 3 (L3) of superficial RSG. Optically stimulating the thalamic axons and terminals in RSC Layer 3 showed

LR neurons to be strongly driven, but RS very little. Morophological studies show that LR and RS neurons' dendrites cluster with their own type.

Interestingly the TC inputs to LR but not RS neurons show synaptic depression. The authors argue that this allows low latency responses, using some detailed modelling of the LR neuron.

In a second experiment, ChR2 injected into claustrum (CLA) and an EYFP tag into AD/AV showed that TC and CLA axons and terminals have distinct and non-overlapping lamination within RSG, such that superficial TC arbors exist most strongly in layers 1a and 3 (where inputs from subiculum also land), while CLA arbors exist in layers 1b/c, 2, and 5. And RS, but not LR, apical dendrites anatomically overlap with CLA arbors in layers 1c and 2. Using the same targeted optogenetic stimulation approach RS neurons have significantly larger responses to CLA input compared to LR neurons (and similarly for inputs from anterior cingulate).

The finding of apparently parallel circuits within the RSG with two different types of input area will be important for understanding the function of the RSG, which is currently receiving much attention.

I have a few questions, mainly for clarification:

1. They argue that facilitating synapses from subiculum to LR neurons would be better for rate coding – as better for overcoming hyperpolarisation from previous spiking. This is easily understood.

It would be good to give an intuitive explanation of why depressing synapses would lead to lower latency responses to head direction inputs from AT – it is one thing to show that a complicated simulation shows the effect and another to explain why it does so.

2. The anatomical demonstrations that TC inputs avoid RS neuron dendrites is somewhat confused by the evidence of such synaptic connections (showing weak facilitation) – maybe this could be explained (there are some but they are weak).

3. Figure 7 implies that TC inputs are to Layer 1a and subicular inputs are to Layer 3, but I don't think this is intended?

---

## [Author Response]

All three reviewers consider that the study addresses important questions in regard to the organization of the neuronal circuits of the granular retrosplenial cortex (RSG). Specifically, the observation that afferent input from the claustrum and the anterior thalamus synapse onto different classes of RSG neurons, is novel and well supported by the present data. However, the reviewers raised a series of concerns that must be addressed before we can reach a decision on the manuscript.While we consider that the points raised by the reviewers must be addressed in full, particular attention should be placed on addressing how the RSG can perform independent spatial and fear computations. Indeed, whereas the connectivity data presented here is suggestive of functional segregation within the proposed circuits, the behavioral and computational implications of these findings are at present highly speculative. The reviewers suggest that a connection between the present findings and circuit computation could be achieved via a model or further experimentation as described in detailed below.

The manuscript has been revised in accordance with the constructive reviewer feedback. In particular, we have extended our circuit mapping experiments to reveal the cell-type specificity of anterior cingulate and dorsal subiculum inputs to superficial retrosplenial cortex (in addition to the inputs from anterior thalamus and claustrum; Figures 2-6). Comprehensive new computational and analytical modeling has also been added. We now show that the observed short-term depression of thalamocortical synapses (Figure 7) can allow low rheobase (LR) neurons in the retrosplenial cortex to encode both rotational head speed and direction, even when the thalamic inputs do not explicitly encode speed (Figures 8-10). We also predict the existence of a clear, explainable tradeoff in encoding of head speed vs direction across individual LR neurons (Figure 10). In addition to identifying the LR cell as the primary initial recipient of spatially-relevant inputs from the anterior thalamus and dorsal subiculum, we have also added more nuanced discussion of the possible functions of cingulate/claustral inputs to regular spiking retrosplenial cells as well as the possible convergence of spatial and non-spatial inputs downstream within the retrosplenial cortex. Detailed point by point responses follow.

Reviewer #1:Brennan et al. present data concerning inputs to layer 1 of retrosplenial cortex arising from the anterior thalamus and claustrum. They provide strong evidence for non-overlap of inputs to retrosplenial cortex from these two structures that segregate both in terms of the specific layer 1 sub-regions and layers III/IV sub-regions that are targeted as well as two different groups of pyramidal cell types that are identified electrophysiologically. The two types are "low-rheobase" (LR) neurons and regular-spiking (RS) neurons. The anatomical and electrophysiological claims are well-evidenced and the data is presented at a time in which there is great interest in retrosplenial cortex and its lamina-specific inputs. There is a nice complement to work concerning inputs from subiculum and hippocampus. The authors also examine to some extent the differential tuning properties of LR and RS neurons in combined modeling/electrophysiology experiments.I find the work interesting, but some of the modeling work seems a bit derivative of work done in a prior publication by the same group (Cell Reports). In addition, the manuscript suffers from an over-interpretation based on the firing properties of anterior thalamic and claustrum neurons. The attribution of claustrum inputs as "higher-order" and "fear-related" goes much too far. Finally, the work lacks an experimental consideration of how the two groups of layer III pyramidal cells are integrated within retrosplenial cortex. For this reason, the emphasis on "parallel circuits" may be misleading. It is not clear if (at the extreme), there are independent circuits in retrosplenial cortex that happen to co-localize to the same region (perhaps to gain input from a third common source) or if the segregation at the level of layer III is followed by integration (for example, by layer V neurons). The combination of anatomy, modeling and electrophysiology has merit, but some of the findings have precedent and, most importantly, the interpretation of the data with respect to the retrosplenial cortex as a whole needs improvement.1. While it is mostly reasonable to speak of anterior thalamic inputs as reflecting head direction inputs (with high input rates), the claustrum inputs cannot be described this way. Unfortunately, there is very little data concerning the response properties of claustrum inputs (though data from O'Mara lends some suggestion of spatial tuning). In addition, a brain region with neurons responding in fear/avoidance based learning scenarios (or for which lesions impair such learning) is all too common. Neurons throughout the brain respond in some way in such conditions as they often do to reward. The authors should refrain from tagging the claustrum inputs in particular with psychology jargon. I realize that this leaves the meaning of the claustrum input as a bit of a mystery, but perhaps other more neuroanatomy or neurophysiology based arguments for the special properties of a claustrum input can be advanced. The authors might also find it much easier to advance arguments of function by putting more focus into the anterior cingulate inputs (cingulate responses are much better defined at this point).

We now present a more nuanced view of the implications of the observed cingulate/claustral targeting of RS but not LR cells. Please see the Discussion sections titled “Downstream targets of LR cells and potential role of RS neurons in retrosplenial spatial computations” and “Potential role of claustral and anterior cingulate inputs” for details of the possible functional implications of the claustral and cingulate inputs to the RSG.

2. Much attention is paid to the layer 1 inputs, but the two input streams diverge equally with respect to layer 3-5. More discussion of the importance of this is desirable and might begin with why each input stream has two very different concentrations of lamina-specific input. What neurons and dendrites are targeted by the layer 3-5 inputs and how does this relate to the layer 1 inputs. In general, the manuscript does not go far enough to incorporate details of circuitry between neurons of retrosplenial cortex. The absence of this leaves the manuscript lacking some originality in terms of data presented and conceptual issues.

Our experiments now consider the deeper inputs using extended CRACM experiments with stimulation sites that range from the midline (layer 1a) through 200 μm into layer 5, encompassing the basal dendrites of the superficial principal neurons that are the focus of this study. We also present more experimental details and additional discussion of within-RSG connectivity. While we have previously reported that LR and RS cells in superficial layers of RSG are unconnected (Brennan et al., 2020), we have now also extended our pair matrix to examine connectivity between layer 3 LR and layer 5 RS cells. Of the 18 connections tested (nine LR→L5RS, nine L5RS→LR), 0 showed connectivity between layer 3 LR cells and layer 5 RS cells. Furthermore, building on the axonal targets we had presented in the first submission, we have examined cross-hemispheric RSG-RSG connectivity using CRACM (Figure 6—figure supplement 1). These experiments show that inputs from contralateral RSG strongly drive RS cells while almost completely avoiding LR neurons. We discuss the potential implications of this connectivity in the Discussion section titled “Downstream targets of LR cells and potential role of RS neurons in retrosplenial spatial computations.”

3. The data presented in figure 4 is compelling with respect to latency to response (5ms is a long time in the brain), but I encourage the authors to spend more time working through the description in results. As written, I think many readers will need to re-read multiple times to understand what is going on (and probably won't re-read). Please make this section more accessible to the broader set of neuroscientists without special expertise in this kind of work.

Following reviewer 2’s suggestion, the entire modeling section has been substantially expanded to focus on head directional coding in LR cells. As such, these new sections have been written with careful focus on clarity, accessibility, and explicit functional implications. Please see response to reviewer 2 major point 3 for full details.

4. Can the authors provide greater context by speaking to the issue of axonal targets of retrosplenial LR and RS neurons?

Please see response to point 2 above and the new Discussion section titled “Downstream targets of LR cells and potential role of RS neurons in retrosplenial spatial computations.”

Reviewer #2:In this manuscript, Brennan et al. expand upon their previous observation of two different pyramidal neuron subtypes in RSG. They had recently shown that RS and LR neurons in superficial RSG exhibit distinct intrinsic excitability profiles and axo-dendritic arborization. Here, they provide evidence that this is paralleled by specific connectivity of long-range inputs and corresponding synaptic dynamics in the two cell types. The authors frame their work through the question of how RSG subserves both fear and spatial coding, concluding that these streams of information are processed by separate and largely independent circuits in RSG built around RS and LR neurons.This is an exciting topic that is potentially a good fit for eLife. How differential connectivity interacts with synaptic and/or intrinsic properties of distinct cell types to shape the computations that contribute to behavioral processes is a pressing problem in the field. And it is highly challenging to address. The specific observation at the core of the manuscript, that thalamic inputs preferentially target LR neurons while claustrum inputs preferentially target RS neurons, is quite interesting. However, the manuscript itself is preliminary and poorly organized, with a number of loose ends and an unclear impact. Substantial revision and expansion are necessary to do justice to the importance of the initial differential connectivity result.1. The authors should finish the incomplete experiments shown in the supplement, which look potentially highly complementary to the data presented in the main figures. Specifically, the authors should complete the subiculum and ACC input anatomy, connectivity, and synaptic dynamics to LR and RS neurons. It is currently a serious missed opportunity. The resultant data will increase the impact of the paper, and it will be particularly useful for model building.

As suggested, we have completed a battery of additional experiments to include full analyses and figures for anterior cingulate and dorsal subiculum inputs to granular retrosplenial cortex. Please see the new Figure 4 (anterior cingulate), Figure 5 (dorsal subiculum), Figure 6 (comprehensive anatomy of each of the 4 input sources, including dual injections and correlations with dendrites), and Figure 7 (synaptic dynamics).

2. The manuscript should be reorganized to present the comprehensive anatomy, connectivity, and synaptic dynamics of AT, CLA, ACC, and SUB in a principled way. Right now, the figures are all over the place, making comparisons challenging. For example, why is there no parallel figure to Figure 3 for CLA inputs? And why are some of the experiments repeated in L5 RS neurons? Is there a parallel system of RS and LR neurons in L5? Have the authors just not found the LR analogues in L5? I am not sure if the L5 data even adds anything – right now it seems like more of a distraction. Maybe it's best to stick to just L2/3.

Each of the 4 input regions (ADAV, CLA, ACC, SUB) examined are now presented using homogenous figure formats (Figures 2-5). We have also included an updated summary figure that directly compares incoming axon-axon, axon-local dendrite, and dendrite-dendrite lamination as Figure 6.

As suggested, we have moved the previous layer 5 results to the supplements (Figure 2—figure supplement 1), in keeping with the reviewer’s suggestion to retain the central focus of the manuscript on the inputs and modeling of superficial RSG neurons.

3. The authors assert (in the abstract, intro, and discussion) that they have identified and modeled how RSG can perform independent spatial and fear computations. They did not test this at all and should not push this agenda without specific, concrete experiments and data that support this direction. What the authors have shown is differential connectivity and synaptic dynamics for two inputs to two cell types. That is basically it. The biophysical model in Figure 4 really has nothing to do with spatial or HD coding. And Figure 7 is not a model at all, but a rudimentary cartoon schematic. In order for this manuscript to have broad impact in the field, the authors should perform a serious modeling effort that integrates what is known about the firing properties of the different neurons that output to RSG with the connectivity and synaptic dynamics they have discovered at RS and LR neurons to test how this system works in vivo. Receptive field properties of RSG neurons have previously been reported by multiple groups (Nitz, Jeffreys, Smith, Bonin, etc): the authors should develop a model that captures some of these in vivo properties and makes non-trivial, testable, and quantitative predictions about, for example, how parallel LR/RS circuitry is beneficial for some aspect of encoding.

We have carried out extensive computational and analytical modeling of thalamic inputs to LR neurons to generate explicitly testable functional predictions about the directional coding capabilities of LR neurons based on the circuitry and dynamics reported in this manuscript. Please see Figures 8-10, and the corresponding new Results sections “Short-term depression of anterior thalamic inputs enables encoding of angular head speed by LR cells”, “Anticipatory firing of thalamic head direction cells improves postsynaptic speed tuning by LR neurons”, and “Non-uniform HD inputs can allow LR cells to encode both head direction and speed, with a tradeoff.” Implications and explicit predictions are then presented in the Discussion section titled “Thalamic inputs to LR cells show short-term depression: implications for angular head velocity and head direction coding in the RSG.”

Reviewer #3:The authors distinguish between two previously characterized types of principal pyramidal neurons in Layer 3 of the granular retrosplenial cortex (RSG): low-rheobase (LR) and regular-spiking (RS) neurons, and identify them by their unique intrinsic properties using patch clamp.Viral injections into the anterodorsal and anteroventral thalamic nuclei (AD/AV) resulted in ChR2-eYFP being expressed in thalamocortical (TC) axons and terminals in layer 1a (L1a; the most superficial part of layer 1) and in layer 3 (L3) of superficial RSG. Optically stimulating the thalamic axons and terminals in RSC Layer 3 showedLR neurons to be strongly driven, but RS very little. Morophological studies show that LR and RS neurons' dendrites cluster with their own type.Interestingly the TC inputs to LR but not RS neurons show synaptic depression. The authors argue that this allows low latency responses, using some detailed modelling of the LR neuron.In a second experiment, ChR2 injected into claustrum (CLA) and an EYFP tag into AD/AV showed that TC and CLA axons and terminals have distinct and non-overlapping lamination within RSG, such that superficial TC arbors exist most strongly in layers 1a and 3 (where inputs from subiculum also land), while CLA arbors exist in layers 1b/c, 2, and 5. And RS, but not LR, apical dendrites anatomically overlap with CLA arbors in layers 1c and 2. Using the same targeted optogenetic stimulation approach RS neurons have significantly larger responses to CLA input compared to LR neurons (and similarly for inputs from anterior cingulate).The finding of apparently parallel circuits within the RSG with two different types of input area will be important for understanding the function of the RSG, which is currently receiving much attention.I have a few questions, mainly for clarification:1. They argue that facilitating synapses from subiculum to LR neurons would be better for rate coding – as better for overcoming hyperpolarisation from previous spiking. This is easily understood.It would be good to give an intuitive explanation of why depressing synapses would lead to lower latency responses to head direction inputs from AT – it is one thing to show that a complicated simulation shows the effect and another to explain why it does so.

The modelling work has been substantially expanded with new insights and predictions about LR function. As such, these new sections have been written with careful focus on clarity, accessibility, and explicit functional implications. Please see response to reviewer 2 major point 3 for full details.

2. The anatomical demonstrations that TC inputs avoid RS neuron dendrites is somewhat confused by the evidence of such synaptic connections (showing weak facilitation) – maybe this could be explained (there are some but they are weak).

We agree with the reviewer that it is best to focus on the short-term dynamics of the synapses that our findings show are prominent and robust (thalamic and dorsal subicular inputs to LR cells; claustrum and cingulate inputs to RS cells). This is especially true with the expanded focus that now includes cingulate and dorsal subiculum and thus increases the number of synapses being examined. Please see the new, hopefully clearer, version of the short-term dynamics figure (Figure 7).

3. Figure 7 implies that TC inputs are to Layer 1a and subicular inputs are to Layer 3, but I don't think this is intended?

The new version of this figure (now Figure 6) has been updated to comprehensively represent laminar inputs into RSG through both the modification of the summary panel and inclusion of incoming axon-axon, axon-local dendrite, and dendrite-dendrite laminar correlations. The data-generated projection density plot shown in panel B (left) now clearly presents the lamination of inputs from all 4 main input regions examined in this manuscript, and this is simplified in the summary circuit cartoon in panel D.